# Combining physical and geochemical methods to investigate lower halocline water formation and modification along the Siberian continental slope

Matthew B. Alkire[1], Igor Polyakov[2], Robert Rember[2], Andrey Pnyushkov[2], Vladimir Ivanov[3, 2], Igor Ashik[3]

[1]Applied Physics Laboratory, University of Washington, Seattle, WA USA
[2]International Arctic Research Center, University of Alaska Fairbanks, Fairbanks, AK USA
[3]Arctic and Antarctic Research Institute, St. Petersburg, RUS

*Correspondence to*: Matthew B. Alkire (malkire@apl.washington.edu)

**Abstract.** A series of cross-slope transects were occupied in 2013 and 2015 that extended eastward from St. Anna Trough to the Lomonosov Ridge. High-resolution physical and chemical observations collected along these transects revealed fronts in the potential temperature and the stable oxygen isotopic ratio ($\delta^{18}$O) that were observed north of Severnaya Zemlya (SZ). Using linear regressions, we describe mixing regimes on either side of the front that characterize a transition from a seasonal halocline to a permanent halocline. This transition describes the formation of lower halocline water (LHW) and the cold halocline layer via a mechanism that has been previously postulated by Rudels et al. (1996). Initial freshening of Atlantic water by sea-ice meltwater occurs west of SZ whereas higher influences of meteoric water and brine result in a transition to a separate mixing regime that alters LHW through mixing with overlying waters and shifts the characteristic temperature-salinity bend from higher ($34.4 \leq S \leq 34.5$) toward lower ($34.2 \leq S \leq 34.3$) salinities. These mixing regimes appear to have been robust since at least 2000.

## 1 Introduction

The role and relative importance of Atlantic water (AW) heat in shaping the Arctic Ocean's ice cover is still under debate (e.g., Polyakov et al., 2012b). One significant source of uncertainty is the impact of diapycnal fluxes on the cold halocline layer (CHL), which separates the fresh and cold surface mixed layer (SML) from the warmer and saltier AW (e.g., Aagaard et al. 1981; Pfirman et al. 1994; Schauer et al. 1997; 2002). The stratification of the CHL, representing strong vertical gradients of salinity and density though a negligible gradient of temperature (resulting in a relatively weak θ-S slope), impedes vertical mixing and upward transport of AW heat (e.g., Rudels et al., 1996; Steele & Boyd, 1998). Underlying the halocline is the reverse thermocline, wherein the temperature increases with depth toward the core of the AW (150-400 m), resulting in a steeper θ-S slope relative to the halocline layer. The LHW is a distinct water mass that is commonly identified

by a "kink" in the θ-S diagram (see Fig. 1c) and forms the base of the CHL; as such, the LHW represents a transition between the halocline and reverse thermocline layers. The LHW was first identified as a separate water mass by Jones and Anderson (1986). They pointed out that the nutrient concentrations were significantly lower than those characterizing the comparatively nutrient-replete upper halocline water of Pacific origin. These differences were further highlighted by the NO parameter, defined as NO = 9*[$NO_3^-$] + [$O_2$], as the LHW was characterized by a local minimum whereas the upper halocline was characterized by a local maximum. We note that some studies interchange the CHL and the LHW. However, we offer the following distinction. While the CHL and LHW may share similar origins/formation mechanisms, we argue that the LHW ($34 \leq S \leq 34.5$) is a comparatively less modified and distinct water mass compared to the CHL ($33 \leq S \leq 34$) that receives inflows from surrounding shelves and is more heavily modified through mixing with overlying waters. The formation of LHW and its modification through diapycnal and/or turbulent mixing with underlying Atlantic water on the Siberian continental slope have important implications for the heat budget and sea ice cover of the Arctic Ocean (e.g., Polyakov et al., 2017). Therefore, it is important to be able to discern between LHW varieties formed by different mechanisms and the modification of these LHW sources through mixing.

Various mechanisms have been proposed for explaining the formation of LHW in the Nansen Basin of the Arctic Ocean. Initially, hypotheses suggested LHW was formed via salinization of Siberian shelf waters through brine rejection and subsequent transport of these waters offshore (Aagaard et al., 1981; Jones & Anderson, 1986; Steele et al., 1995). Such hypotheses have been previously referred to as the "*advective mechanism*" of LHW formation in the literature due to its formation entirely on the shelves and subsequent advection into the deep basins. At present, it is generally agreed that the primary mechanism of LHW formation results from the modification of AW by melting sea ice upon entry into the Arctic through Fram Strait and the Barents Sea (Rudels et al., 1996; 2004). In this scenario, relatively fresh ($34 \leq S \leq 34.3$) SML water undergoes convective mixing through cooling and brine release during winter sea ice formation. This winter mixed layer (WML) is advected along the Siberian continental margin and is eventually capped by low-salinity shelf waters moving offshore, limiting the depth of subsequent convection. This hypothesis has been typically referred to as the "*convective mechanism*" of LHW formation in the literature. We point out that the "*advective*" and "*convective*" labels for differentiating LHW formation are misleading, particularly since the latter mechanism depends upon the advection of the WML eastward along the slope until low-salinity shelf waters are advected offshore and increase the stratification. Convective and advective processes are involved in both formation mechanisms; therefore, we have chosen to replace these terms with "*basin-derived*" and "*shelf-derived*", respectively, to minimize further confusion.

Steele and Boyd (1998) suggested an "*advective-convective mechanism*" wherein the CHL/LHW is derived from both salty shelf waters originating from the Kara and (primarily) Barents Seas (i.e., "*shelf-derived*") and the WML of the deep Nansen Basin where convective mixing homogenizes surface waters that have been previously freshened by sea ice meltwater (*"basin-derived"*). The salty shelf waters advect northward into either a winter mixed layer (100-150 m thick) of similar salinity (S~34) or below a summer mixed layer and into a seasonal halocline layer that will be eroded during convective mixing the following winter. This combined mixed layer will eventually progress eastward where fresher shelf

waters from the eastern Kara and Laptev Seas will mix into surface and near-surface waters, providing the necessary stratification to cap the LHW against deeper convective mixing and form a permanent halocline layer. In our view, the "capping" process is primarily responsible for the formation of the CHL atop the LHW that is formed by either *shelf-derived* and/or *basin-derived* processes. Rudels et al. (2004) also suggested that both mechanisms of halocline formation (i.e., *shelf-*

and *basin-derived*) are possible, resulting in two different sources of halocline water in the eastern Arctic: Fram Strait Branch (FSB) and Barents Sea Branch (BSB) halocline waters. According to Rudels et al. (2004), the FSB branch variety of halocline water is formed via interaction between inflowing AW and sea ice north of Svalbard and subsequent convection in the Nansen Basin, quite similar to the *basin-derived LHW* of Rudels et al. (1996). The BSB variety is formed in the Barents Sea through a complex combination of processes (including cooling, melting sea ice, mixing with freshwater from the

Norwegian Coastal Current, net precipitation, river runoff from the Kara Sea, and brine release during ice formation, though the latter process is thought to be a less likely component) resembling the mechanism outlined by Steele & Boyd (1998). Rudels et al. (2004) further postulates that after entering the Eurasian Basin through St. Anna Trough (SAT), the BSB halocline water remains close to the Siberian continental slope, and after crossing the Lomonosov Ridge ventilates the lower halocline of the Makarov Basin between the Mendeleyev Ridge and the Chukchi Cap as well as the southern Canada Basin.

In contrast, the FSB halocline water is displaced farther offshore, ventilating the halocline of the Amundsen and Makarov Basins, as well as northern Canada Basin.

The BSB halocline water has been found to be saltier, thicker, and warmer compared to colder and fresher FSB halocline waters. These distinctions can be visually recognized in a θ-S diagram: the cooler FSB variety is expected to exhibit a sharp θ-S kink close to the freezing point whereas the warmer BSB variety is generally characterized by a smoother

kink farther from the freezing point line. Thus, differences can be observed in the properties of halocline waters occupying the slope ("on-slope") versus those located farther offshore ("off-slope"). Woodgate et al. (2001) attributed these cross-slope distinctions to differences in the formation processes (i.e., *shelf-* vs. *basin*-derived halocline water). Rudels et al. (2004) attributed the higher salinity of the BSB halocline water to lower Atlantic water temperatures in the Barents Sea since cooler waters will melt less ice. The higher temperature of BSB halocline water was attributed to enhanced turbulent mixing

between the BSB halocline water and underlying (and warm) AW as they are advected eastward along the Siberian slope. They argued that the mixing acts to entrain more AW into the halocline, making it both thicker and warmer while simultaneously cooling the AW layer. Dmitrenko et al. (2011) argued that turbulent vertical mixing occurring locally on the Laptev Sea slope explains the differences observed between warmer/on-slope and cooler/off-slope LHW properties observed along a regularly occupied section (~126 °E) in the Laptev Sea between 2002 and 2009; however, they did not consider the

possibility of lateral advection of cross-slope differences from upstream.

Despite the importance of river water and sea-ice melt/brine in LHW formation, few studies have utilized $\delta^{18}$O to investigate the formation or modification of LHW through mixing. It is the purpose of this paper to pair a large number of closely spaced $\delta^{18}$O measurements (focused on the halocline layer) with CTD-based temperature and salinity measurements

collected along a series of cross-slope transects extending from the SAT to the Lomonosov Ridge to improve our understanding of LHW formation, circulation, and modification through mixing with Siberian shelf waters and underlying AW.

## 2 Data & methods

In collaboration with the Arctic and Antarctic Research Institute (St. Petersburg, Russia), oceanographic cruises were conducted within the Eurasian Basin and along the slope of the Kara, Laptev, and East Siberian Seas during summers of 2013 (August 23-September 19) and 2015 (August 28-September 26) aboard the research vessels *Akademik Fedorov* and *Akademik Tryoshnikov*, respectively (Fig. 2). Totals of 116 (2013) and 94 (2015) hydrographic stations were occupied during the cruises. At all stations, a rosette equipped with 24 Niskin bottles, a Seabird SBE9*plus* CTD (conductivity-temperature-depth), and additional sensors were deployed (further details provided in Supplementary Text S1). At all but 8 (2013) and 6 (2015) stations, water samples were collected for a variety of chemical and biological measurements at routine depths of 500, 250, 200, 150, 140, 130, 120, 110, 100, 90, 80, 70, 60, 50, 40, 30, 20, 10, and 2-4 m (surface).

Samples for $\delta^{18}O$ analyses were collected into 20 mL glass vials, the caps of which were fitted with conical polyethylene inserts, parafilmed, and shipped to the Stable Isotope Laboratory, Oregon State University, for analysis via the $CO_2$ equilibration method on a Finnegan Mat 251 mass spectrometer. Totals of 1254 and 1940 samples were collected in 2013 and 2015, respectively. Precision was estimated to be ± 0.02 ‰ (2013) and 0.04 ‰ (2015), based on the mean standard deviations of field duplicates. Laboratory duplicates were also conducted to ascertain the performance of the mass spectrometer. Of these, the mean standard deviation was ± 0.02 ‰ during both years. Bottle salinities are not reported due to malfunction of the salinometer available aboard each ship. Instead, CTD properties were matched to bottles via averaging measurements associated with each bottle trip depth using the bottle (.ros) files recorded for each cast. The accuracy of temperature and conductivity measurements recorded by the CTD is expected to be within ± 0.001°C and ± 0.0003 S m$^{-1}$, respectively, per manufacturer specifications. For further details and data access, readers are referred to the Supplementary Materials, NABOS project website (http://research.iarc.uaf.edu/NABOS2/), and the NSF Arctic Data Center (https://arcticdata.io).

## 3 Results

Transects occupied during 2013 indicated that the base of the WML, identified as a potential temperature minimum ($\theta_{min}$) below the warmer and fresher SML (Rudels et al., 1996), was associated with salinities > 34. The presence of a seasonal, rather than a permanent, halocline layer was evidenced by relatively weak stratification between the base of the WML and the θ-S bend identifying LHW (Fig. 3), potential temperatures near the freezing point at S = 34.1 (e.g., red lines in Fig. 3d),

and higher salinities (S ≥ 34) at 40-50 m depth (Fig. 4d); thus, a permanent halocline was either very weak or absent throughout most of our study area (Steele and Boyd, 1998; Kikuchi et al., 2004; Bourgain and Gascard, 2011).

At stations in the western part of the study area, it was also apparent that the θ-S kink was sharp, close to the freezing point, and at a relatively shallow depth (typically ≤ 50 m) (Fig. 3a-c) indicating that the halocline was *basin-derived* and likely seasonal (Steele et al., 1995; Rudels et al., 1996; Steele and Boyd, 1998). Farther eastward, the L3 and L4 transects exhibited a front that separated stations closer to shore versus those farther offshore (Fig 3d-e). This front marked a significant change in the core AW temperature (Fig. 4f) as well as a θ increase (Fig. 4e) and $\delta^{18}O$ decrease (Fig. 4c) in the salinity range 34.4 ≤ S ≤ 34.5 and an apparent shift of the θ-S bend marking the position of LHW towards lower salinities (34.2 ≤ S ≤ 34.3) (e.g., Fig. 3d). Coincident with this θ-S front, there was also a change in the predominant source of freshwater near the surface. Sea-ice meltwater (SIM) fractions were positive and larger than fractions of meteoric water (MW) along the lengths of sections SAT, L1, and L2 as well as the nearshore stations comprising sections L3 and L4; however, transects L5, L5.5, and L6 all exhibited predominate freshening by MW (Fig. 4a-b). Bauch et al. (2014) reported a similar, zonal gradient along the Siberian slope, with increasing contributions of both MW and brine from west to east, where shelf waters are advected offshore at ~140 °E (in the northeastern Laptev Sea) and contribute to layers overlying LHW (S ≤ 33).

The easternmost stations of the SAT transect and the southernmost stations of transects L2 and L3 exhibited θ-S characteristics expected for BSB AW (black lines in Fig. 3a, c, d). At L5, three stations inshore of the ~1250 m isobath (< 77.2 °N) exhibited θ-S characteristics (Fig. 3f) synonymous with northern Barents Sea Shelf Water (Woodgate et al., 2001). These observations generally agree with the expectation that BSB waters are restricted to the slope and indicate that the predominance of FSB (or *basin-derived*) LHW throughout most of the study area. We note that the θ-S characteristics of BSB waters were not apparent along transects L1 or L4, possibly indicating that we failed to sample far enough inshore to capture BSB waters at these transects.

## 4 Discussion

### 4.1 Geochemical separation of mixing regimes

The coincident shift in freshwater sources was also marked by an obvious change (or "break") in the $\delta^{18}O$-S slope at 34.4 ≤ S ≤ 34.5 (Fig. 5a). A change in $\delta^{18}O$-S slope may indicate a change in the mixing regime that typically involves the introduction of a new water mass. For example, on the western side of the front, the salinity-$\delta^{18}O$ data may be explained by simple mixing between the Atlantic layer and a SML that is freshened predominately by SIM. The change in $\delta^{18}O$-S slope at

the front indicates the introduction of MW as the primary source of freshwater (Fig. 4a-b). However, it is unclear from the data presented in Fig. 5a whether or not mixing of MW is restricted to shallower depths (associated with salinities < 34.5; i.e., *to the left of the slope break*) or if this new mixing regime extends over the full salinity range (i.e., both right and left of the slope break). Therefore, we explore this change in mixing in more detail by comparing simple, linear regressions of
salinity and $\delta^{18}$O. At each transect, two groups of regressions were assessed. The first group included data *to the right of the slope break* (S $\geq$ 34.5). The second group included data *to the left of the slope break* (34 $\leq$ S < 34.5).

First, we report results of the linear regressions encompassing the higher salinity data (S $\geq$ 34.5). The stations occupied along the SAT, L1, L2, and southern portions of the L3 and L4 transects (including those stations exhibiting BSB influence) all exhibited similar slopes (i.e., linear mixing regimes) in $\delta^{18}$O-S space that indicated predominate freshening by SIM
(Table 1). This freshening by SIM was evident by the higher SIM fractions observed at these stations (e.g., Fig. 4b) as well as the range (between -4.7 and -8.9 ‰) of intercepts (S = 0) computed from simple, linear regressions of the data (Table 1). Data collected from this group of stations all appeared to plot along a single, linear mixing line at the higher end of the salinity range (S $\geq$ 34.5). In fact, separate linear regressions from these transects were all statistically indistinguishable (Table 1); thus, a single $\delta^{18}$O-S linear regression was constructed using these data to define what we refer to as the "SIM
mixing branch" for S $\geq$ 34.5 (Fig. 5b). Similarly, data collected from stations farther offshore on the L3 and L4 transects were combined with those along the L5 transect to construct the "MW mixing branch" for S $\geq$ 34.5 (Fig. 5c and Table 1). Notably, the slopes and intercepts characterizing the mixing regimes of the SIM and MW branches were significantly different for S $\geq$ 34.5. This difference indicates that the shift in mixing that occurred across the SZ front was not restricted to lower salinities (S $\leq$ 34.5) but extended to higher salinities.

Next, we report the results of linear regressions conducted on data to the left of the slope break, specifically in the salinity range typically associated with LHW (34 $\leq$ S < 34.5). Linear regressions conducted on data collected from the stations comprising the SIM branch returned coefficients (Table 2) that were statistically indistinguishable from the more saline (S $\geq$ 34.5) regressions (Table 1); thus, the SIM branch extended over most of the water column at stations west of SZ. In contrast, there were significant changes in the $\delta^{18}$O-S slopes characterizing the stations of the MW branch (Fig. 5c). Linear
regressions returned steeper slopes and more negative intercepts that indicated higher influences of both MW and brine (i.e., negative SIM); a situation typical of Laptev Sea shelf waters (Bauch et al, 2011). Net ice formation (freezing) results in the rejection of salts from the sea ice matrix as well as a small, but preferential rejection of $^{16}$O over $^{18}$O. As a result, brine is characterized by higher salinities and more negative $\delta^{18}$O values whereas sea ice (and therefore sea ice meltwater) is characterized by lower salinities and somewhat more positive $\delta^{18}$O values. This fractionation is not large; fractionation
factors range between about 1.6 and 2.8 ‰ depending upon the age of the ice and the rate of freezing (Macdonald et al., 1995; Melling and Moore, 1995; Eicken, 1998), but it does deflect a simple, linear mixing line between AW and MW to the right, resulting in a steeper salinity-$\delta^{18}$O slope (as illustrated in Fig. 5e). Therefore, the change in mixing regime across the

SZ front altered the $\delta^{18}$O-S slopes of both the lower (S $\geq$ 34.5) and upper (34 $\leq$ S < 34.5) portions of the water column. We will discuss potential mechanisms to explain these changes in section 4.2.

Eastward of ~126 °E, stations along the L5.5 and L6 transects generally exhibited $\delta^{18}$O values that were somewhat higher/more positive compared to the linear regression/mixing line defined for the lower salinity range (34 $\leq$ S < 34.5) of the MW branch (Fig. 5d & Table 2). Thus, this mixing relationship is altered between the Laptev and East Siberian Seas, perhaps due to a larger influence from positive (or less negative) SIM and/or entrainment of thermocline waters containing a larger influence from AW. Rivers flowing into the East Siberian Sea are typically characterized by more negative $\delta^{18}$O values compared to the Lena, Ob, and Yenisey Rivers (Cooper et al., 2008) so increased MW influence cannot solely explain the more positive $\delta^{18}$O values. Sea-ice meltwater influences are generally higher/more positive in the East Siberian Sea compared to the Laptev Sea, as the Laptev is characterized by net sea-ice formation over melting (and thus a net negative SIM contribution), even during summer months (Bauch et al. 2011; 2013; Anderson et al., 2013). There are fewer data from the higher salinity range (S $\geq$ 34.5) to assess differences in $\delta^{18}$O-S slopes between the MW branch and transects L5.5 and L6; however, the available data suggest little-to-no statistically significant differences in the regression coefficients (Table 1), indicating that changes in mixing were likely driven by surface and near-surface mixing (i.e., larger contributions from positive SIM).

**4.2 Interpretation of mixing branches:** *basin-derived* **vs.** *shelf-derived*

Aksenov et al. (2011) describe the Arctic Shelf Break Branch (ASBB) of the Arctic Circumpolar Boundary Current as a narrow current that transports halocline waters from the Barents and Kara Seas northward via the SAT and eastward along the Siberian continental slope over approximately the 1500 m isobath. Their description is similar to the circulation scheme of *shelf-derived* (or BSB) LHW proposed by Rudels et al. (2004). More recently, Bauch et al. (2016) used a combination of geochemical tracers collected across the Siberian continental margin between 2005 and 2009 in a principle components analysis to identify four separate LHW types: c1 (S~33), c2 (S~34), c3 (S~34.2), and c4 (S~34.4). Types c2 and c4 were the most commonly observed, originating at the shelf break north of SZ (type c4) or ~126 °E (type c2) and both extending eastward to at least ~140 °E. Bauch et al. (2016) argued that the regular presence of type c4 LHW north of SZ suggests the Kara Sea as a source of this LHW type. They further postulated that this water leaves the Kara Sea via SAT and/or Voronin Trough and circulates around the slope via the ASBB. Similarly, they argue that type c2 LHW is formed in either the northwestern Laptev Sea or (more likely) in the southeastern Kara Sea and transported to the slope via Vilkitsky Strait.

The description offered by Bauch et al. (2016) for the formation and circulation of LHW types c2 and c4 is also reminiscent of *shelf-derived,* BSB LHW. However, these LHW types are found both on and off the slope, rather than restricted to the continental slope as expected for BSB LHW (Woodgate et al., 2001; Rudels et al., 2004). Bauch et al. (2016) argue that off-slope transport might occur directly or via recirculating waters from the eastern Eurasian Basin

(Rutgers van der Loeff et al., 2012). We observed θ and $\delta^{18}O$ characteristics associated with salinities of 34, 34.2, and 34.4 that are quite similar to the LHW types described by Bauch et al. (2016); however, these similarities were restricted to MW branch stations located offshore of the continental slope. In addition, the $\delta^{18}O$ values associated with salinities 34.4-34.5 at SIM branch stations were much higher than those reported by Bauch et al. (2016). These apparent discrepancies suggest
different formation and/or circulation schemes compared to those provided by Bauch et al. (2016). Here, we offer an alternative hypothesis.

The WML observed at stations located in the western transects (SAT, L1, and L2) is formed through freshening of AW with SIM and some small contribution of MW (likely from net precipitation and runoff entering the Barents Sea) to establish a seasonal halocline; these processes produce the SIM branch. However, this branch only represents an initial
condition, as further stratification is necessary to prevent winter mixing from eroding the LHW (i.e., lower salinity waters from the Laptev Sea shelf are needed to "cap" the LHW) and the SIM branch is not observed eastward of SZ. Therefore, the SIM branch is synonymous with the seasonal halocline and the front observed north of SZ marks the start of the transition from the seasonal to the permanent halocline.

We interpret the transition from SIM to MW branches north of SZ as descriptive of the formation of *basin-derived* LHW
(Rudels et al., 1996). We suggest that this transformation occurs via homogenization of the upper water column through mixing and salinization from brine expulsion during sea ice formation. To test this hypothesis, we estimated new mixed layer salinities at the SIM branch stations assuming mixing penetrated to the previous WML depth and then calculated the changes in salinity and $\delta^{18}O$ due to sea ice formation. The mean WML depth and salinity was ~50 m and 34.37, respectively, for all SIM branch stations (see Supplementary Table S1). Mixing of the water columns at individual stations
down to their respective WMLs resulted in a new, mean mixed layer salinity of ~33.83 with a corresponding $\delta^{18}O$ value of ~0 ‰ (calculated using the SIM branch regression). Too few $\delta^{18}O$ data were collected from the near surface to directly calculate new mixed layer $\delta^{18}O$ values. The use of the SIM branch regression to estimate the new mixed layer $\delta^{18}O$ likely underestimates the influence of MW to the surface layer, particular at the front (sections L3 and L4). As a result, the final $\delta^{18}O$ values computed after sea ice growth are likely biased slightly high/more positive. Brine expulsion from 1.0-1.5 m of
sea ice growth increases the salinity to between 34.38 and 34.66 and decreases $\delta^{18}O$ to between -0.05 and -0.08 ‰. These resulting salinity and $\delta^{18}O$ values roughly plot along the upper or lower MW branches (Fig. 5e & Supplementary Figure S1). Therefore, mixing with overlying, less saline waters results in small changes to salinity and $\delta^{18}O$ that are sufficient to initiate a movement from the SIM mixing relationship (prevalent on the western side of the defined front) to the MW mixing relationship (prevalent on the eastern side of the defined front) in salinity-$\delta^{18}O$ space that also corresponds with the
migration of the θ-S "kink" (or "bend") that has typically been used to identify LHW. Continued influence from Siberian shelf waters caps the LHW, isolating it from subsequent surface mixing, and results in a break in the $\delta^{18}O$-S slope that

defines a shallower mixing regime characterized by a steeper $\delta^{18}$O-S slope and highly negative intercept (i.e., the lower MW branch). We also argue that this latter process is responsible for the formation of the CHL.

While mixing down to the previous year's WML (or shallower) might be expected given the increase in freshwater inventories (and stratification) moving from west to east along the slope, deeper mixing was observed in the study region between 2013 and 2015 (Polyakov et al., 2017). The depth of the 34.4 isohaline ranged between 60 and 100 m at the MW branch stations. If we consider mixing down to 100 m and 1 m of ice formation, the resulting salinity (34.50) and $\delta^{18}$O (0.07 ‰) resemble the upper MW branch at the break point. Thus, both shallower (~60 m) and deeper (~100 m) mixing result in a transition from the SIM branch to the MW branch. Although mixing and brine release can account for salinity and $\delta^{18}$O changes, additional mixing (either lateral or vertical) with warm AW is needed to produce the $\theta \approx$ -1 °C that is associated with the LHW of the MW branch. A mixture comprising ~79 % of newly formed MW branch water (34.38, -0.08 ‰, and -1.89 °C) and ~21 % AW (34.9, 0.3 ‰, and 2 °C) would produce the salinity (34.4), $\theta$ (-1.07 °C), and $\delta^{18}$O (0 ‰) observed. We have included this simple mixing scenario to further test the possibility of our proposed mechanism to explain both the $\delta^{18}$O and potential temperature observations in the LHW. While we do not claim that this simple mixing is necessarily responsible for the observed halocline water properties, we note that such mixing can explain our observations.

It is also important to note that MW must have been supplied to the region north of SZ to define the front separating SIM and MW branches. We adopt the suggestion made by Bauch et al. (2016) that waters moving off the shelf in the northeastern Laptev Sea (i.e., along the Lomonosov Ridge) are recirculated westward, except we suggest this recirculation does not necessarily provide four distinct sources of halocline water. Any shelf contribution with a salinity exceeding that of the relatively fresh polar mixed layer will contribute to the halocline. Our observations suggest that the majority of these shelf contributions will occur eastward of the SZ front. We argue that LHW (34.2 < S < 34.5) is primarily *basin-derived* and initial shelf water contributions serve to cap LHW (and begin to establish the permanent halocline) whereas further contributions to the halocline will have a salinity < 34.2 and therefore contribute to the "lower MW mixing branch" and build the CHL. In support of this hypothesis, we note that the salinity and $\delta^{18}$O values characterizing the four LHW types defined by Bauch et al. (2016) form a salinity-$\delta^{18}$O mixing line ($\delta^{18}$O = 0.9828S – 33.901) similar to the lower MW branch identified in this study (Supplementary Figure S2). This could indicate that the four LHW types described by Bauch et al. (2016) are actually mixtures of *basin-derived* LHW and increasing contributions of MW progressing eastward from SZ.

### 4.3 Stability of $\delta^{18}$O-S mixing regimes

Using salinity and $\delta^{18}$O observations, we have outlined a hypothesis to describe the transition from a seasonal halocline, formed due to mixing between AW and SIM west of SZ, to a permanent halocline involving winter mixing, ice formation, and the introduction of Siberian shelf waters characterized by high influences of MW and brine east of SZ that largely follows the hypothesis previously described by Rudels et al. (1996). However, we have thus far relied upon data collected

during a single summer (2013). How robust are the $\delta^{18}$O-S mixing relationships we have defined using the 2013 data? In this section, we conduct similar linear regressions using data sets collected by numerous projects over a period of > 15 years.

As noted in section 2, a second cruise was conducted in 2015 that re-occupied some of the transects surveyed in 2013 (i.e., SAT, L1, L5, and L6). The 2015 data suggest a very similar hydrographic setting as that encountered in 2013 (i.e.,

weak/absent CHL with similar cross-slope fronts observed at repeated transects). The salinity-$\delta^{18}$O data generally agree with the scheme proposed here (see Supplementary Tables S2 & S3) as they plot along the three branches characterized using the 2013 data (Fig. 6a). For example, the regression coefficients computed using data collected from transects SAT and L1 in 2015 are very similar to those defining the SIM branch. Similarly, regression coefficients computed from data collected along transects L5 and L6 in 2015 closely resemble the upper (S $\geq$ 34.5) and lower (34 $\leq$ S < 34.5) MW branches. The

similarity in $\delta^{18}$O-S slopes and intercepts along these transects suggest similar processes are responsible for the transition between the SIM and MW mixing regimes and that the location of the front marking this transition likely occurred in a similar region (i.e., between transects L1 and L5, in the vicinity of SZ).

We further test the stability of these $\delta^{18}$O-S mixing regimes by estimating linear regressions for these two salinity ranges using data collected as part of the North Pole Environmental Observatory (NPEO) (Alkire et al., 2015), Leg 2 of the ARK-

XXII expedition (Bauch et al., 2011), and from the O-18 Atlas (Schmidt et al., 1999). The locations of the water samples collected during these three projects are shown in Fig. 2. The O-18 Atlas data were restricted to a latitude range of 75-90ºN and longitude range of 65-160ºE to more closely match the general area (Siberian shelves and the Nansen, Amundsen, and Makarov Basins) surveyed during the 2013 and 2015 cruises. The salinity-$\delta^{18}$O data collected by each of these three programs all generally plot along the three mixing lines defined using the 2013 data (Fig. 6b-d). Furthermore, the regression

coefficients from the upper (S > 34.5) and lower (34 < S < 34.5) salinity ranges were quite similar to those characterizing the MW branches, with the exception of the ARK-XXII expedition (Supplementary Table S4). The slope and intercept derived from the ARK-XXII data resembled the SIM mixing branch; however, a restriction of these data to the longitude range 110-160°E resulted in regression coefficients that more closely resembled the MW branch. Thus, these comparisons generally confirm the apparent dominance of the MW branch east of ~110°E (approximate position of the L3 transect) and restricted

nature of the SIM branch. The similarity of the regression coefficients estimated from the NPEO (observations collected between 2000 and 2015), ARK-XXII (2007), and O-18 Atlas (1967-2008) data sets with those estimated from the 2013 and 2015 cruises further suggests that these mixing relationships have been relatively stable since at least 2000.

## 5 Summary & conclusions

A front was observed north of SZ at sections L3 and L4 that separated mixing branches dominated by either SIM (west of

the front) or MW (east of the front). We interpret observations of salinity-$\delta^{18}$O regressions as indicative of two stages of mixing that contribute to the formation of *basin-derived* LHW. The first stage is described by the SIM branch as AW is

freshened predominately by ice melt and is then subject to further modification through subsequent vertical mixing (with less saline, overlying waters) and ice formation. The vertical mixing reduces both salinity and $\delta^{18}O$ of the WML and ice formation then increases the salinity but only slightly decreases the $\delta^{18}O$ (see Fig. 5e). This process results in a shift from the SIM branch to the MW branch north of SZ and causes a prominent break in the salinity-$\delta^{18}O$ slope at $34.4 \leq S \leq 34.5$.

The second stage is described by mixing with Siberian shelf waters containing large influences from MW and brine (negative SIM) that isolates the LHW from surface processes and builds the CHL, resulting in another change in $\delta^{18}O$-salinity slope. Farther east at transects L5.5 and L6, stations generally plotted along the MW branch but exhibited signs of additional modification that are likely a consequence of mixing with East Siberian Sea shelf waters that contain larger influences from sea-ice meltwater (positive SIM). A comparison of these results with recent studies raises questions as to

whether the LHW types identified by Bauch et al. (2016) are independent, advective sources of LHW or products of mixing between *basin-derived* LHW and less saline shelf waters. Additional observations are necessary to further address these distinctions.

We also note that colder waters originating from the Barents Sea were generally found at stations inshore of the ~1600 m isobath (in agreement with Aksenov et al., 2011) along transects L1 and L2 whereas stations farther offshore were either

clearly dominated by warmer, FSB AW or exhibited mixing between the warmer FSB and colder BSB waters. However, no such fronts occurred in $\delta^{18}O$-S (all stations plotted along the SIM branch). At the L5 section, three stations inshore of the ~1250 m isobath (< 77.2 °N) exhibited BSB-like $\theta$-S characteristics but anomalously low $\delta^{18}O$ values ($\leq$ -0.2 ‰) between salinities 34.4 and 34.7, indicating large contributions of brine. All other stations on L5 plotted along the MW branch. Thus, if BSB LHW was advected within the ASBB, it was restricted to the shallowest depths encountered during the 2013

and 2015 cruises and likely underwent additional modification through interaction with shelf waters. Thus, *basin-derived*, FSB LHW was the dominant LHW variety observed throughout most of our study area.

Finally, comparisons against other data sets collected across the Eurasian Basin of the Arctic Ocean (see Fig. 6) suggest that the salinity-$\delta^{18}O$ mixing regimes defined here have remained relatively stable despite changes to the sea ice cover (Polyakov et al., 2017), the temperature and volume of AW inflow (e.g., Polyakov et al., 2012a), and distribution of river

runoff (Guay et al., 2001; Dmitrenko et al., 2005) for > 15 years. The apparent, robust nature of the salinity-$\delta^{18}O$ mixing regimes suggests that the processes responsible for LHW formation and modification have not been greatly altered by these important environmental changes, perhaps due to seasonal processes such as river discharge and sea-ice melting and freezing that may be delayed or diverted but not otherwise impacted by these changes. Instead, we speculate that such changes might alter the position of the front(s) marking the transition between the SIM and MW branches and/or result in data plotting in

different positions along the established mixing lines (e.g., closer to or farther away from the AW endmember in salinity-$\delta^{18}O$ space). Thus, while the distribution and/or strength of stratification provided by the halocline in certain regions (e.g., Amundsen Basin) is altered by such changes, the processes responsible for halocline water formation remain consistent.

This implies that salinity-$\delta^{18}O$ relationships may be a more reliable method for characterizing halocline water formation and mixing during periods of significant variability.

**Data availability**

All data presented and/or described in this manuscript can be accessed via the National Science Foundation Arctic Data
Center (https://arcticdata.io) via the following six data and metadata sets:

Igor Polyakov. 2016. NABOS - CTD Survey Data 2013. Arctic Data Center. doi:10.18739/A2M37F.

Igor Polyakov. 2016. NABOS - Water Quality and Physical Oceanography Data from the Eastern Eurasian and Makarov
Basins, and Northern Laptev and East Siberian Seas in 2013. Arctic Data Center. doi:10.18739/A2G95H.

Igor Polyakov. 2016. NABOS - Chemistry Data 2013. Arctic Data Center. doi:10.18739/A2QS91.

Igor Polyakov. 2016. NABOS II - Water Quality and Physical Oceanography Data from the Eastern Eurasian and Makarov
Basins, and Northern Laptev and East Siberian Seas in 2013 - 2015. Arctic Data Center. doi:10.18739/A20955.

Igor Polyakov. 2016. NABOS II - CTD Survey Data 2015. Arctic Data Center. doi:10.18739/A2436Q.

Igor Polyakov. 2016. NABOS II - Chemistry Data 2015. Arctic Data Center. doi:10.18739/A27S9P.

**Competing interests**

The authors declare that they have no competing interests

**Acknowledgements**

Alkire acknowledges funding from the National Science Foundation (PLR-1203146 AM003) and the National Oceanic & Atmospheric Administration (NA15OAR4310156).  Alkire would also like to thank Dr.'s Michael Steele and Rebecca
Woodgate (Applied Physics Laboratory) for helpful comments and suggestions during the preparation of this manuscript as well as Wendy Ermold for help in figure preparation.  Ivanov acknowledges funding from the Ministry of Education and Science of the Russian Federation (project RFMEFI61617X0076).

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

**Tables**

Table 1. Linear regression analyses (restricted to salinities ≥ 34.5) of salinity-$\delta^{18}$O measurements collected along transects
occupied during the 2013. Slopes, intercepts, correlation coefficients (r) and associated standard errors (se) are reported for each
transect as well as the collection of transects comprising the sea-ice melt (SIM) and meteoric (MW) water branches.

| Transect | Slope | se | Intercept | se | Corrcoeff | Stations |
|----------|-------|-----|-----------|------|-----------|----------|
| SAT | 0.2059 | 0.0395 | -6.9306 | 1.3715 | 0.6016 | 109-116 |
| L1 | 0.2626 | 0.0545 | -8.8868 | 1.8945 | 0.8005 | 97-108 |
| L2 | 0.2471 | 0.0292 | -8.3596 | 1.0156 | 0.6656 | 82-91 |
| L3 upper | 0.2477 | 0.0373 | -8.4049 | 1.2958 | 0.6919 | 76-81 |
| L4 upper | 0.1415 | 0.0412 | -4.7276 | 1.4298 | 0.6632 | 68-69 |
| | | | | | | |
| **SIM Branch** | **0.2287** | **0.0347** | **-7.7306** | **1.2044** | **0.6632** | |
| | | | | | | |
| L3 lower | 0.4589 | 0.0424 | -15.7646 | 1.4703 | 0.8864 | 70-75 |
| L4 lower | 0.5693 | 0.0577 | -19.6058 | 2.0035 | 0.8776 | 63-66 |
| L5 | 0.631 | 0.0379 | -21.793 | 1.3131 | 0.8690 | 10-26 & 60-62 |
| | | | | | | |
| **MW Branch** | **0.6016** | **0.0321** | **-20.7517** | **1.1141** | **0.8328** | |
| | | | | | | |
| L5.5 | 0.7521 | 0.2182 | -25.9928 | 7.5479 | 0.7054 | 45-59 |
| L6 | 0.7265 | 0.0924 | -25.0783 | 3.195 | 0.8537 | 29-38 |

**Table 2.** Linear regression analyses (restricted to the salinity range: $34 \leq S < 34.5$) of salinity-$\delta^{18}O$ measurements collected along transects occupied during the 2013. Slopes, intercepts, correlation coefficients (r) and associated standard errors (se) are reported for each transect as well as the collection of transects comprising the sea-ice melt (SIM) and meteoric (MW) water branches.

| Transect | Slope | se | Intercept | se | Corrcoeff | Stations |
|---|---|---|---|---|---|---|
| SAT | 0.2361 | 0.0522 | -7.9857 | 1.786 | 0.6863 | 109-116 |
| L1 | 0.1113 | 0.225 | -3.6912 | 7.7323 | 0.1266 | 97-108 |
| L2 | 0.2048 | 0.063 | -6.9215 | 2.1638 | 0.5105 | 82-91 |
| L3 upper | 0.1975 | 0.0605 | -6.684 | 2.0747 | 0.6446 | 76-81 |
| L4 upper | 0.1816 | 0.0659 | -6.1461 | 2.2606 | 0.7216 | 68-69 |
| | | | | | | |
| **SIM Branch** | **0.1871** | **0.0422** | **-6.3148** | **1.4463** | **0.6167** | |
| | | | | | | |
| L3 lower | 1.4715 | 0.0455 | -50.6961 | 1.5618 | 0.9837 | 70-75 |
| L4 lower | 1.1334 | 0.1018 | -39.0734 | 3.4945 | 0.9345 | 63-66 |
| L5 | 1.2739 | 0.0462 | -43.9486 | 1.5859 | 0.9356 | 10-26 & 60-62 |
| | | | | | | |
| **MW Branch** | **1.3126** | **0.0364** | **-45.2639** | **1.2482** | **0.9421** | |
| | | | | | | |
| L5.5 | 0.6543 | 0.0822 | -22.5928 | 2.8197 | 0.7411 | 45-59 |
| L6 | 0.643 | 0.075 | -22.2101 | 2.5717 | 0.8867 | 29-38 |

**Figures**

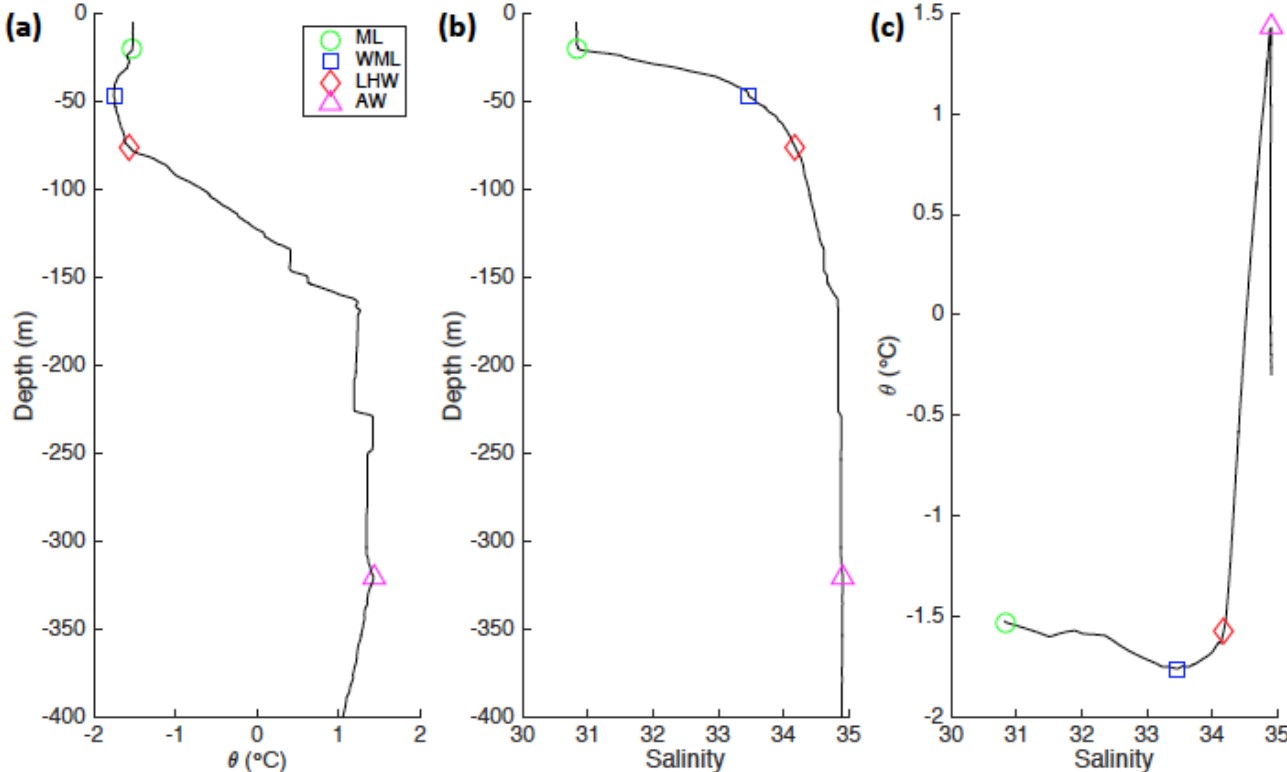

Figure 1. Vertical profiles of (a) potential temperature (θ) and (b) salinity, as well as the corresponding θ-S diagram (c) for a single station (station 26) occupied in 2013. The bottom boundaries of the surface mixed layer (SML) and winter mixed layer (WML) are shown by the green circles and blue squares, respectively. The θ-S bend (or "kink") that has been typically used to identify the position of lower halocline water (LHW) is shown by the red diamonds. The $\theta_{max}$ marking the core of the Atlantic water layer (AW) is shown by the magenta triangles. The halocline is the layer between the SML and LHW. The reverse thermocline is the layer between the LHW and AW. The base of the WML was determined as the $\theta_{min}$ below the SML. The LHW position was computed via the method outlined in Bourgain and Gascard (2011).

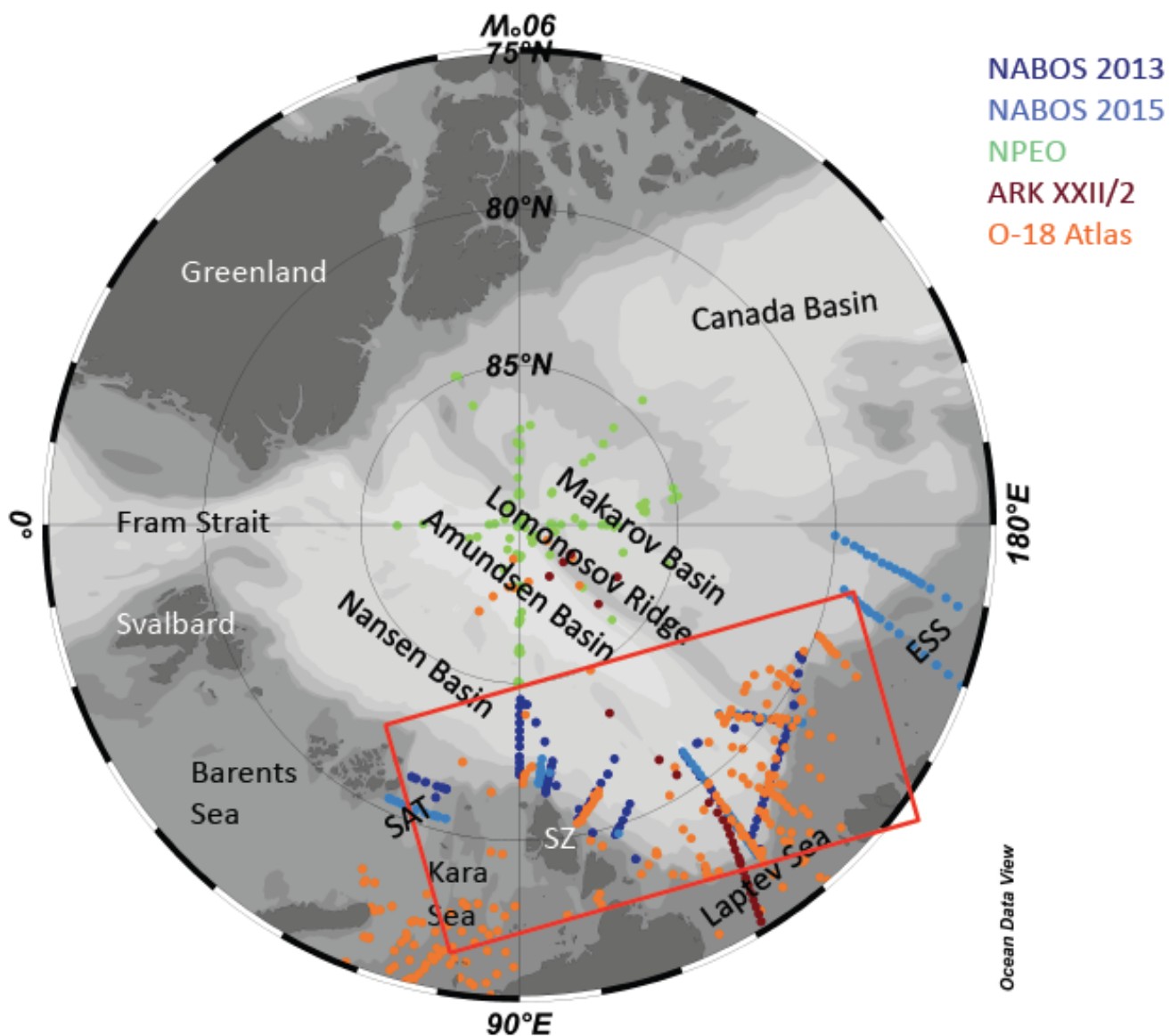

**Figure 2. General map of Arctic Ocean showing study area (red box) and stations occupied during 2013 cruise (dark blue circles) and 2015 cruise (light blue circles) as well as stations occupied as part of the North Pole Environmental Observatory (green circles), ARK XXII/2 expedition (dark red circles), and O-18 Atlas (orange circles). SAT = St. Anna Trough; SZ = Severnaya Zemlya; ESS = East Siberian Sea. The map were created using Ocean Data View software (version 4.7.6) (Schlitzer, 2016).**

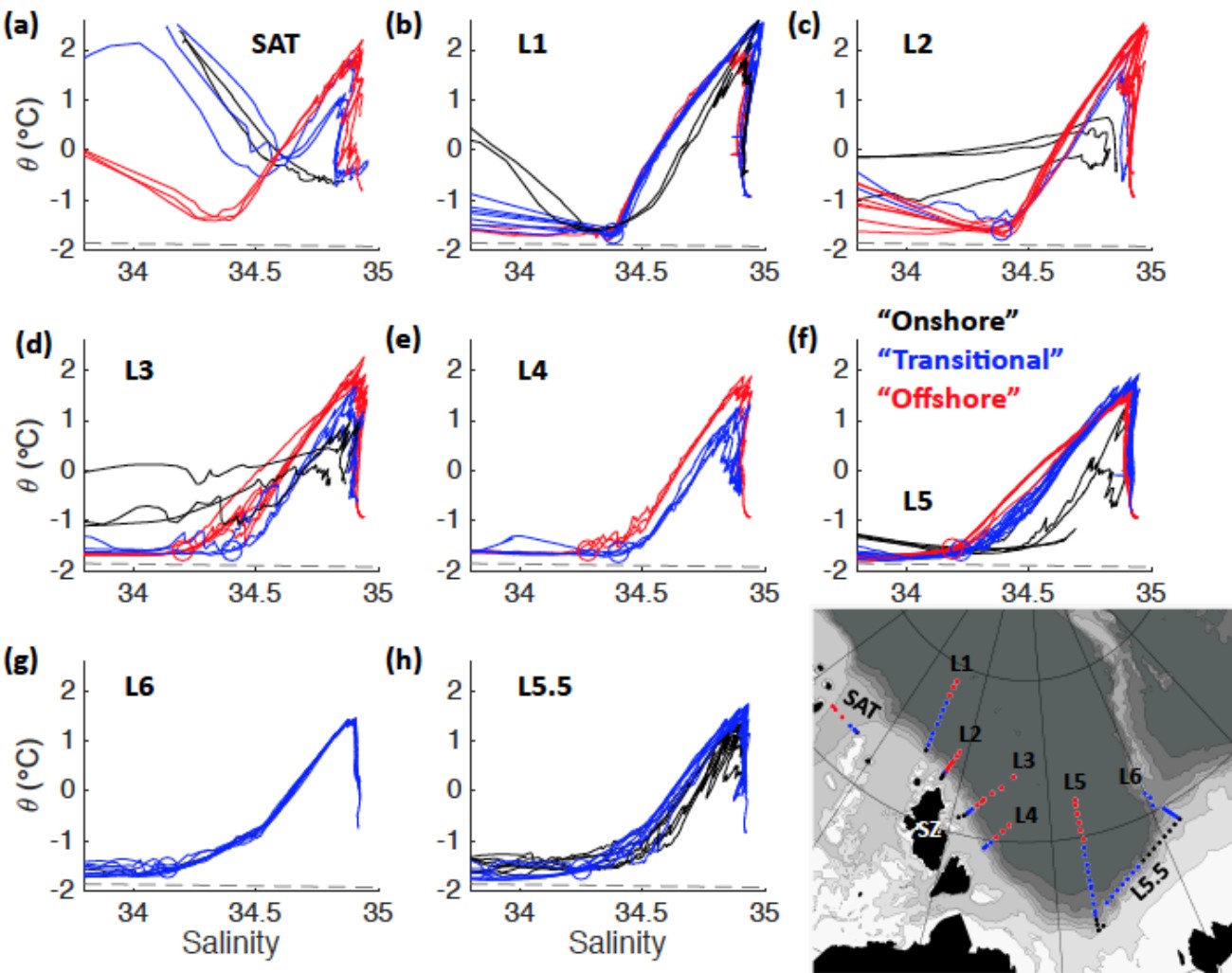

**Figure 3.** The panels exhibit θ-S diagrams for all data collected during the 2013 cruise. Data are divided among subpanels according to transect (SAT, L1, L2, L3, L4, L5, L5.5, and L6) with the locations of each transect shown in the inset map. The θ-S data measured at each station are colored black (closest to shore or "onshore"), blue ("transitional" between onshore and offshore), or red (farthest "offshore") according to its relative onshore vs. offshore position. Along the St. Anna Trough (SAT) section, the colors indicate the relative position of stations farthest west (red), central/east (blue), and farthest east/shallow (black) rather than onshore/offshore. The relative positions were defined differently along each transect according to fronts observed in θ-S characteristics as described in the text. Red and blue circles on these diagrams show the mean positions of LHW at the transitional and offshore stations along each transect, respectively. LHW positions along L1 and L2 did not significantly differ between transitional and offshore stations; therefore, only a single position is plotted. Note that all stations on the L6 transect were plotted in blue as there was little difference among stations indicative of a θ-S front.

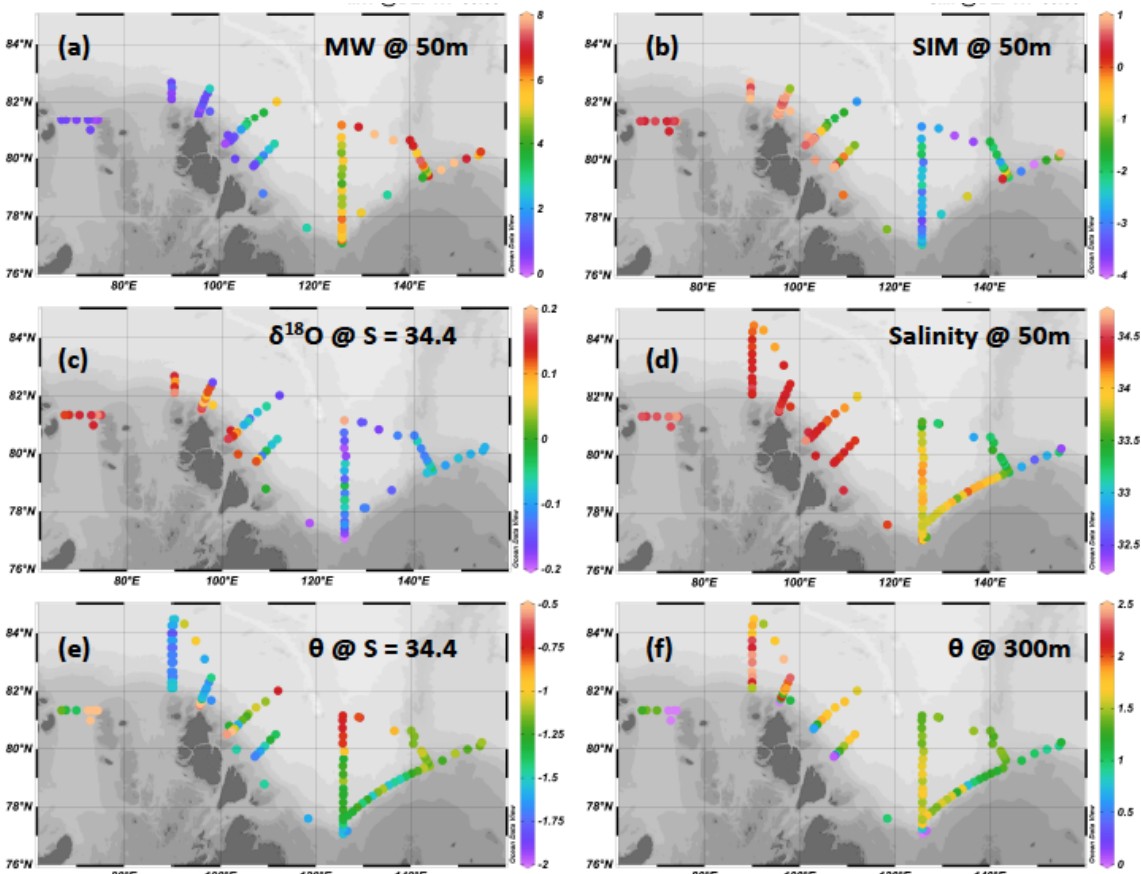

**Figure 4.** Maps of the (a) meteoric water (MW) fraction (%) at 50 m depth, (b) net sea-ice meltwater (SIM) fraction (%) at 50 m depth, (c) $\delta^{18}O$ (‰) on the 34.4 isohaline, (d) salinity at 50 m depth, (e) potential temperature (ºC) on the 34.4 isohaline, and (f) potential temperature (ºC) at 300 m (i.e., the approximate depth of the Atlantic water core). The MW and SIM fractions were calculated using a coupled water type analysis conserving salinity, $\delta^{18}O$, and mass according to methods outlined in Alkire et al. (2015); specific details regarding the methods of the analyses are provided in the Supplementary Text S2. Maps were created using Ocean Data View software (version 4.7.6) (Schlitzer, 2016).

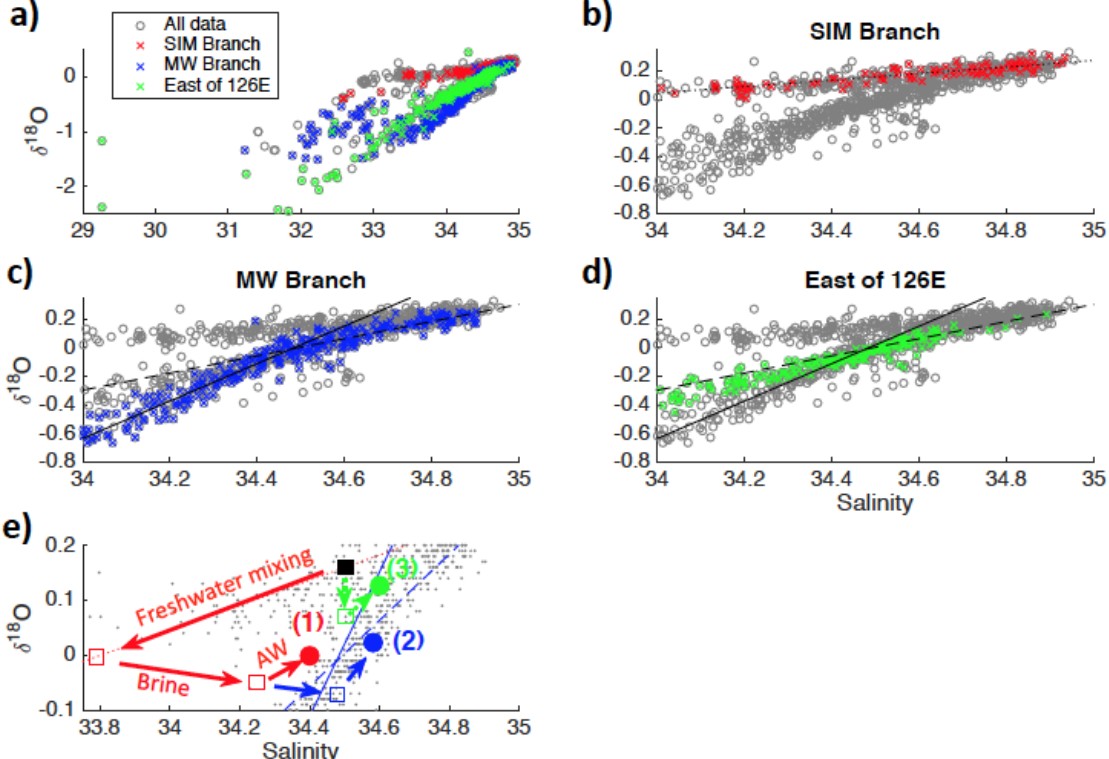

**Figure 5.** Plots of salinity versus the stable oxygen isotopic ratio ($\delta^{18}$O) measured during the 2013 cruise. The entire data set is plotted in each panel as gray circles. Data collected from stations comprising the sea-ice meltwater (SIM) branch, meteoric water (MW) branch, and remaining stations located east of the L5 transect (L5.5 and L6 transects) are plotted as red, blue, and green x's in panels (b), (c), and (d), respectively. Linear regressions characterizing the SIM ($\delta^{18}$O = 0.2287*S – 7.7306; $R^2$ = 0.44) and MW ($\delta^{18}$O = 0.6016*S – 20.7517; $R^2$ = 0.69) branches (S $\geq$ 34.5) are plotted as dotted and dashed lines, respectively. The lower MW branch (34 $\leq$ S < 34.5) is plotted as a solid line ($\delta^{18}$O = 1.3126*S – 45.2639; $R^2$ = 0.89). Both MW branches are plotted in panel (d) for comparison against data along L5.5 and L6 transects. Note that the inclusion of all data collected east of 126°E results in a linear regression that was statistically indistinguishable from the MW branch ($\delta^{18}$O = 0.63S – 21.8; $R^2$ = 0.71); however, this was not the case for the lower salinity range; thus, these stations were excluded in the definition of the MW branches. Panel (e) illustrates the transition from the SIM branch to the MW branch via mixing with overlying freshwaters, salinization through sea ice formation/brine release, and mixing with Atlantic waters (AW). The red pathway illustrates the effect of vertical mixing down to ~50 m (the mean winter mixed layer depth at SIM branch stations), brine expulsion due to the formation of 1 m of sea ice, and mixing with AW in a 21:79 ratio to form lower halocline water with a salinity of 34.4 and $\delta^{18}$O of 0 ‰ (1). The blue pathway deviates from the red pathway due to additional ice formation (1.5 m instead of 1 m) to form lower halocline water with a salinity of 34.58 and $\delta^{18}$O of 0.02 ‰ (2). The green pathway illustrates the effect of vertical mixing to 100 m, 1 m of sea ice formation, and AW mixing to form lower halocline water with a salinity of 34.6 and $\delta^{18}$O of 0.13 ‰ (3). Empty squares indicate transition points after each step whereas filled circles indicate the final halocline water product formed by the three potential pathways. All three pathways yield salinity and $\delta^{18}$O combinations near (but not directly on) the MW mixing branches, indicating some additional processes and/or mixing (such as freshwater influence from river runoff) takes place during the transition from the SIM branch to the MW branch. A larger version of this figure is available in the Supplementary Information, Figure S1.

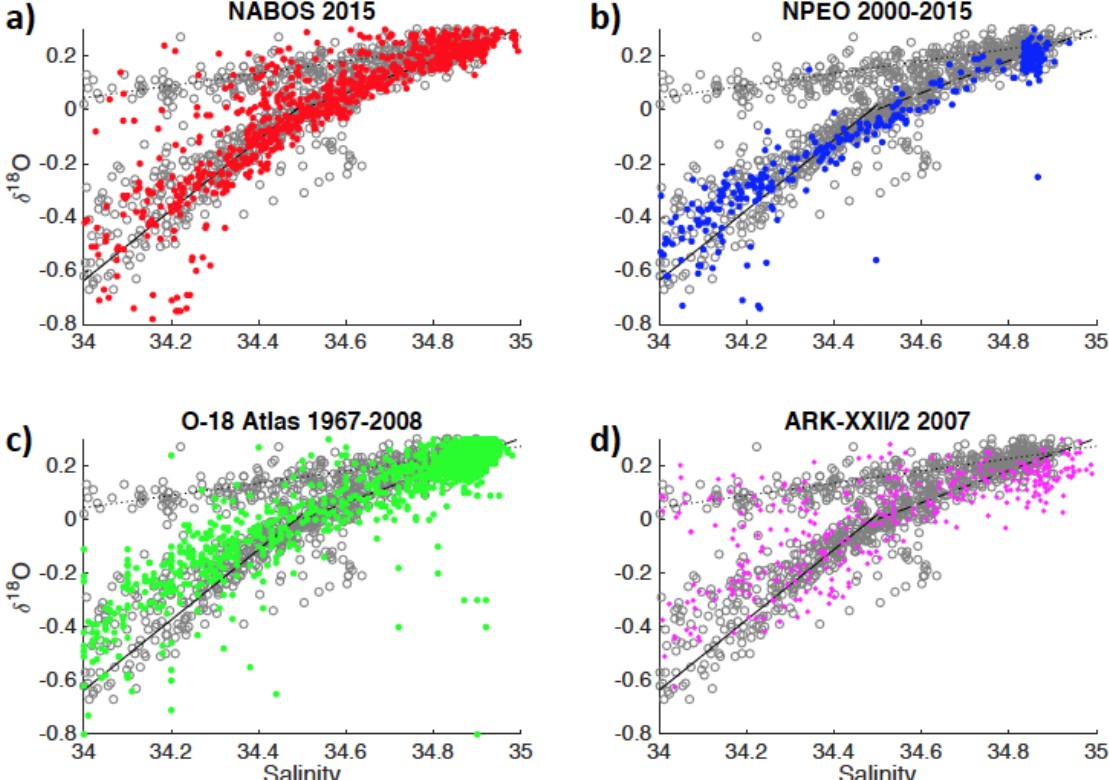

**Figure 6. Comparison of data and linear regressions defining the SIM, MW, and lower MW branches defined during the 2013 cruise against additional data sets collected within the study region and in the deep basins of the eastern Arctic (Nansen, Amundsen, and/or Makarov Basins): (a) 2015 cruise; (b) North Pole Environmental Observatory (NPEO); (c) Oxygen-18 Database; and (d) ARK-XXII/2 expedition. In each panel, the 2013 data are plotted as gray circles and the linear regressions are plotted as dotted (SIM Branch), dashed (MW Branch), and solid (lower MW branch) lines. Data from each of the four cruises are plotted as (a) red, (b) blue, (c) green, and (d) magenta dots to indicate the general correspondence of these data with the mixing regimes defined by the three branches. Station locations corresponding to each data set are shown in Fig. 2. The NPEO data was previously published by Alkire et al. (2015) and can be accessed online at the NSF Arctic Data Center (https://arcticdata.io). The 2015 NABOS cruise data can be accessed online at the NSF Arctic Data Center. Data from the Oxygen-18 Database (Schmidt et al., 1999) were restricted to longitudes 65-160 °E and latitudes 75-90 °N to closely resemble the area sampled for this study. The data can be accessed online at https://data.giss.nasa.gov/o18data/. Data from the ARK-XXII/2 cruise aboard the *Polarstern* were published by Bauch et al. (2011) and can be accessed online via PANGEA (doi:10.1594/PANGAEA.763451).**