# Peer review of "Combining physical and geochemical methods to investigate lower halocline water formation and modification along the Siberian continental slope"

_Ocean Science, 2017_

## Referee Comment (RC1) · Anonymous Referee #1 · 31 Jul 2017

This manuscript examines the importance of river water and sea-ice melt/brine in lower halocline water (LHW) formation through mixing. The study is based on observations/sampling in the Eurasian basin of the Arctic Ocean and along the continental slope of the Kara, Laptev, and East Siberian Seas during summers of 2013 and 2015. The study, which uses $\delta18O$ along with CTD measurements, suggests that LHW is formed by convective mechanisms with two stages of convective mixing during the transit along the continental slope and thus offers an alternative hypothesis to the study published by Bauch et al. (2016). The Alkire study makes an important contribution

to the understanding of the mechanism which may impact the vertical heat fluxes between the (seasonally) ice covered surface layer and the underlying "warm" Atlantic water (AW). The paper is concise, well structured, and well written.

The main issue that needs to be addressed is the result presented in the last sentence of the abstract. The authors postulate that: "These mixing regimes appear to have been robust since at least 2000". This means that the mixing regimes are stable although the region experienced significant changes in sea ice cover and temperature/volume of AW during the last two decades (references in the manuscript). Does that also mean that the intensity of vertical mixing between LHW and AW has not changed since 2000? Considering the importance of these conclusions I'm not convinced that they are a result of the analysis presented in this paper. The authors state that the conclusion was drawn based on a comparison against other data sets (?) collected between 2000 and 2015 (page 8, line 14). This description seems too vague to me. It should be discussed in more detail.

Technical corrections:

The citation "Janout et al. (2015)" from the list of references is missing in the text/figures.

The citation "Rudels et al. (1994)" from the list of references is missing in the text/figures. Is it Guay et al 2001 (text) or 2011 (references)?

The map shown in figure 1 is to small (at least for me). Maybe it would be better to show a map of the entire Arctic Ocean with the research area highlighted. Because this is the first map presented in the manuscript you should add longitude and latitude.

---

## Referee Comment (RC2) · Anonymous Referee #2 · 23 Aug 2017

This manuscript addresses the Arctic Ocean halocline, its formation and evolution in the eastern Nansen and Amundsen basins. By the use of oxygen-18 data in combination with standard ctd measurements a front in the halocline characteristics is identified north of Severnaya Zemlya. West of the front sea ice melt water dominates in the halocline, while east of the front a stronger presence of meteoric water (runoff & precipitation) is observed. The stronger influence of meteoric water is due to vertical mixing with overlying, less saline water, but I am not sure if the authors also claim that the presence of meteoric water indicates major different sources of the halocline water.

[Figure]

Instead of discuss this issue here, I will first go through the manuscript from the top and then return to this question below.

Specific points: Page 1,lines 26-27: The halocline is not just the "kink" but also includes the water of different TS slope (stronger salinity change) up to the level of the seasonal deepening in winter, if lucky to be identified by a temperature minimum. It should perhaps also be stated that the lower halocline water was introduced by Jones and Anderson (1986) to distinguish it from the nutrient rich upper halocline with salinity around 33.1.

Page 2, lines 6-8: In this scenario the upper layer is formed by melting sea ice and mixing the melt water into the upper part of the Atlantic water creating a fairly saline, cold upper layer. As the upper layer is advected eastward seasonal sea ice melt creates a summer halocline, which is removed by ice formation and brine release in winter, creating a winter mixed layer above the Atlantic water. AS a general remark, I think that the use of an advected halocline versus a convective halocline complicates the picture. The winter mixed layer is advected and homogenized until it becomes covered by an outflow of less saline shelf water. If the shelf water is more saline and denser than the surface layer, it is injected into the water column below the upper layer. In both cases advection as well as convection are involved.

Page 2, lines 11-13: As this description stands, it does not differ from the mechanism described in the paragraph above. As I interpret the schematics in Steele and Boyd (1998) the advective contribution comes from the northern Barents Sea and the northern Kara Sea.

Page 2, lines 13-16: The description of the process proposed by Kikuchi et al. (2004) cannot be correct. Freezing on the Atlantic water at its entrance, should that be possible, would lead to convection and homogenization of the Atlantic layer and perhaps convection to the bottom. A low salinity upper layer must exist for this mechanism to work, creating the bent TS curves.

Page 2, lines 18-23: Rudels et al. (2004) claim that the water from the Barents Sea that eventually contributes to the Barents Sea branch halocline water also is formed by sea ice melting on Atlantic water. The higher salinity compared to the Fram Strait branch halocline water is due to the lower temperature of the Atlantic water when it encounters the sea ice in the eastern Barents Sea.

Page 2, line 29: The Barents Sea branch halocline is initially more saline and eventually it also becomes warmer and thicker due to mixing with underlying Atlantic water.

Page 4, first paragraph: There is no halocline water mass in the Nansen Basin west of Severnaya Zemlya and the thermocline and halocline coincide.

Page 4, line 9: In the supplementary material the depth of the winter mixed layer with salinity >34 ranges between 30m to 94m (L1) and between 42m and 58m (L2). Most of the depths were larger than 50m. (There were also two stations with depths close to 200m on L2 but the temperatures were close to 0C and the salinities >34.8. Should these observations be correct we would have winter mixing into the Atlantic water. This shows that the temperature minimum as a limit for winter convection should be used with care. However, in general it underestimates the winter mixed layer depth.)

Page 5, lines 19-21: Freezing shifts the mixing line between Atlantic water and meteoric water to the right, making it steeper. The Laptev Sea shelf input would then contain more brine than the East Siberian Sea input.

Page 5, last paragraph: Here I am not sure that I follow the authors thinking. As the water from the Nansen Basin (the sea ice melt water branch) becomes covered by les saline shelf water with larger content of meteoric water, vertical mixing with the overlying water will lead to a mixing line with the observed slope. However, the bulk of the halocline water mass is derived from the winter mixed layer advected towards the Laptev Sea. The mixing changes slightly the properties of the halocline, but it does not provide any significant volume. Is this what the authors mean, or do they claim that here exits a major different source of the halocline water containing initially more

meteoric water?

Page 6, lines 10-14: Once the low salinity polar mixed layer is formed, any shelf contribution having higher salinity than the polar mixed layer will contribute to the halocline. The question is, how large are these contributions compared with the initial Fram Strait branch and Barents Sea branch contributions?

Page 6, lines 26-28: This description is essentially correct, but why not state explicitly that to create an halocline water mass from the winter mixed layer, this layer has to be capped by a water mass with lower salinity advected from the Laptev Sea shelf?

Page 6, lines 30-32, Page 7, lines 1-5: These mixing examples are interesting, but I think that it would be simpler to think of the winter mixed layer being capped by low salinity water. The initial thermostad and halostad would then be freshened by mixing with the low salinity layer above and heated and become more saline by mixing with the Atlantic water below. This would create the vertical gradients in temperature and salinity that characterize the halocline. Furthermore, I thought that freezing does not fractionate the oxygen isotopes. Brine rejection would then not change the oxygen-18 value. What is the reference for the adopted slope? On page 8, line 13 brine rejection is characterized as "negative sea ice melt". Would that not give a slope that increases the delta-oxygen-18 value as well as the salinity?

Page 7, lines 6-7: To me the lower halocline water and the cold halocline layer are the same, at least in this location. Once we enter the Canada Basin with Pacific inflow through Bering Strait the situation is different.

Page 7, lines 23-24: I agree with this statement.

Page 8, lines 10-11: Why does the delta-oxygen-18 decreases? A reference is needed.

Page 8, lines 17-21: Does this mean that there are two separate branches of halocline water, or only that when the Fram Strait and the Barents branches cross the front, their upper parts are changed from being homogenized by brine rejection and convection to

[Figure]

evolve by mixing with overlying low salinity and underlying Atlantic water?

Figure 1: I would have appreciated to see not only TS curves but also profiles from the different sections.

Summary: My concern about this manuscript should be clear after the comments above. If the authors mean that there are two distinct branches, one sea ice melt branch and one meteoric water branch, they have to argue their case better. If it is only a transition between the two branches across the front, the manuscript does not bring much new information. However, the use of the oxygen-18 data is interesting, but some of the adopted slopes have to be explained better.

The manuscript might be published after major revisions, some based on my comments given above.

---

## Author Response (AR1)

**Combining physical and geochemical methods to investigate lower halocline water formation and modification along the Siberian continental slope**

Matthew B. Alkire[1], Igor Polyakov[2], Robert Rember[2], Andrey Pnyushkov[2], Vladimir Ivanov[3, 2], Igor Ashik[3]

[1]Applied Physics Laboratory, University of Washington, Seattle, WA USA
[2]International Arctic Research Center, University of Alaska Fairbanks, Fairbanks, AK USA
[3]Arctic and Antarctic Research Institute, St. Petersburg, RUS

*Correspondence to*: Matthew B. Alkire (malkire@apl.washington.edu)

**Responses to Reviewer Comments**

**We thank the two reviewers for their helpful suggestions and comments regarding this manuscript. As a result of these comments, we have undertaken significant revisions of the paper, both in the revision and addition of figures and a significant expansion of the text to further clarify major discussion points. We have also moved some tables from the Supplementary Information to the main text, in support of this expanded discussion.**

**We have addressed each of the comments provided by the two reviewers. Our responses are located directly below each specific comment, in bold type. Page and line numbers mentioned in our responses refer to a revised version of the manuscript that we have prepared and are ready to submit for consideration.**

**Anonymous Referee #1**

This manuscript examines the importance of river water and sea-ice melt/brine in lower halocline water (LHW) formation through mixing. The study is based on observations/sampling in the Eurasian basin of the Arctic Ocean and along the continental slope of the Kara, Laptev, and East Siberian Seas during summers of 2013 and 2015. The study, which uses δ18O along with CTD measurements, suggests that LHW is formed by convective mechanisms with two stages of convective mixing during the transit along the continental slope and thus offers an alternative hypothesis to the study published by Bauch et al. (2016). The Alkire study makes an important contribution to the understanding of the mechanism which may impact the vertical heat fluxes between the (seasonally) ice covered surface layer and the underlying "warm" Atlantic water (AW). The paper is concise, well structured, and well written.

**Thank you.**

The main issue that needs to be addressed is the result presented in the last sentence of the abstract. The authors postulate that: "These mixing regimes appear to have been

robust since at least 2000". This means that the mixing regimes are stable although the region experienced significant changes in sea ice cover and temperature/volume of AW during the last two decades (references in the manuscript). Does that also mean that the intensity of vertical mixing between LHW and AW has not changed since 2000?

**No, this does not mean that the intensity of vertical mixing between LHW and AW has not changed since 2000.  We apologize for the confusion and thank the reviewer for pointing out this obscurity.  We argue that the apparent consistency of the mixing regimes, as defined by the salinity-d18O mixing lines, indicate that the processes responsible for the formation of halocline waters has remained more-or-less constant regardless of the large and important environmental changes observed over the Arctic Ocean during the past two decades.  Although the spatial distribution and strength of stratification provided by the halocline may have been altered, the processes responsible for halocline water formation have apparently not changed.**

**We have added a new section (4.3) as well as an additional paragraph at the end of the paper (Page 11, Lines 24-32) that provides further context supporting our conclusion.**

Considering the importance of these conclusions I'm not convinced that they are a result of the analysis presented in this paper. The authors state that the conclusion was drawn based on a comparison against other data sets (?) collected between 2000 and 2015 (page 8, line 14). This description seems too vague to me. It should be discussed in more detail.

**The data sets collected between 2000 and 2015 are those presented in Figure 6 of the revised manuscript.  They include data collected in 2013 and 2015 (presented in this study) as well as from the North Pole Environmental Observatory (2000-2015), ARK-XXII/2 (2007), and O-18 Atlas (1967-2008).  The locations of observations collected during these programs are shown in Figure 2 of the revised manuscript.**

**We have added a section (4.3) to the revised manuscript that discusses the comparison in further detail.**

Technical corrections:
The citation "Janout et al. (2015)" from the list of references is missing in the text/figures.

**This citation has been deleted from the list of references.**

The citation "Rudels et al. (1994)" from the list of references is missing in the text/figures.

**This citation has been deleted from the list of references.**

Is it Guay et al 2001 (text) or 2011 (references)?

**The citation in the text is correct (2001) and has been corrected in the references.**

The map shown in figure 1 is to small (at least for me). Maybe it would be better to show a map of the entire Arctic Ocean with the research area highlighted. Because this is the first map presented in the manuscript you should add longitude and latitude.

**A larger map has been added as a new figure (Figure 2) in the revised manuscript.**

**Anonymous Referee #2**

This manuscript addresses the Arctic Ocean halocline, its formation and evolution in the eastern Nansen and Amundsen basins. By the use of oxygen-18 data in combination with standard ctd measurements a front in the halocline characteristics is identified north of Severnaya Zemlya. West of the front sea ice melt water dominates in the halocline, while east of the front a stronger presence of meteoric water (runoff & precipitation) is observed. The stronger influence of meteoric water is due to vertical mixing with overlying, less saline water, but I am not sure if the authors also claim that the presence of meteoric water indicates major different sources of the halocline water.

**We do not claim that the presence of meteoric water indicates major differences in the source(s) of halocline water. Instead, we utilize the available data to generally confirm the "convective mechanism" of halocline water formation that was previously postulated by Rudels et al. (1996). The front indicates an important geographic region where the "seasonal halocline" begins a transformation to the "permanent halocline".**

**We have significantly expanded the discussion in sections 4.1 and 4.2 and added text to the Abstract (Page 1, Lines 13-15) to further clarify this point.**

Instead of discuss this issue here, I will first go through the manuscript from the top and then return to this question below.

Specific points: Page 1, lines 26-27: The halocline is not just the "kink" but also includes the water of different TS slope (stronger salinity change) up to the level of the seasonal deepening in winter, if lucky to be identified by a temperature minimum. It should perhaps also be stated that the lower halocline water was introduced by Jones and Anderson (1986) to distinguish it from the nutrient rich upper halocline with salinity around 33.1.

**We have revised the text to better define the halocline as suggested by the reviewer and have included a citation for Jones and Anderson (1986). These additions can be found between Pages 1 (Line 26) and 2 (Line 9) of the revised manuscript.**

Page 2, lines 6-8: In this scenario the upper layer is formed by melting sea ice and mixing the melt water into the upper part of the Atlantic water creating a fairly saline, cold upper layer. As the upper layer is advected eastward seasonal sea ice melt creates a summer halocline, which is removed by ice formation and brine release in winter, creating a winter mixed layer above the Atlantic water. As a general remark, I think that the use of an advected halocline versus a convective halocline complicates the picture. The winter mixed layer is advected and homogenized until it becomes covered by an outflow of less saline shelf water. If the shelf water is more saline and denser than the surface layer, it is injected into the water column below the upper layer. In both cases advection as well as convection are involved.

**The reviewer makes an excellent point and we have added text to the manuscript pointing out this problem with the typical "advective" and "convective" labels of halocline water formation presented in the scientific literature (Page 2, Lines 16-28) and offer the terms of "shelf-derived" and "basin-derived" halocline waters instead.**

Page 2, lines 11-13: As this description stands, it does not differ from the mechanism described in the paragraph above. As I interpret the schematics in Steele and Boyd (1998) the advective contribution comes from the northern Barents Sea and the northern Kara Sea.

**We note that, while Figure 9 in Steele and Boyd (1998) does indeed illustrate salty shelf waters advected from both the Barents and Kara Seas, in the text they write that, "The only salty (i.e., S > 33) shelf sea in the eastern longitudes of the Arctic Ocean is the Barents Sea, which led *Aagaard et al.* [1981] to speculate that this would be the most likely source of CHL formation via the mechanism show in Figure 2a."**

**We have included both the Kara and Barents Seas as possible sources of relatively salty shelf water in our discussion of Steele and Boyd's mechanism but note that the Barents Sea is likely the primary source.**

**The text has been corrected and expanded in the revised manuscript (between Page 2, Line 29 and Page 3, Line 4) to offer a better description of the "advective-convective" mechanism described in Steele and Boyd (1998).**

Page 2, lines 13-16: The description of the process proposed by Kikuchi et al. (2004) cannot be correct. Freezing on the Atlantic water at its entrance, should that be possible, would lead to convection and homogenization of the Atlantic layer and perhaps convection to the bottom. A low salinity upper layer must exist for this mechanism to work, creating the bent TS curves.

**We have removed the lines referring to the Kikuchi et al. (2004) paper to avoid further confusion. The "ideal" situation introduced by Kikuchi et al. is not mentioned further in the manuscript and not relevant to the discussion.**

Page 2, lines 18-23: Rudels et al. (2004) claim that the water from the Barents Sea that eventually contributes to the Barents Sea branch halocline water also is formed by sea ice melting on Atlantic water. The higher salinity compared to the Fram Strait branch halocline water is due to the lower temperature of the Atlantic water when it encounters the sea ice in the eastern Barents Sea.

**We have added statements pointing out this distinction on Page 3, Lines 23-26 of the revised manuscript.**

Page 2, line 29: The Barents Sea branch halocline is initially more saline and eventually it also becomes warmer and thicker due to mixing with underlying Atlantic water.

**We have added statements pointing out this distinction on Page 3, Lines 23-26 of the revised manuscript.**

Page 4, first paragraph: There is no halocline water mass in the Nansen Basin west of Severnaya Zemlya and the thermocline and halocline coincide.

**This paragraph refers to the study area as a whole and indicates a weak or absent halocline. We later argue that there exists a seasonal halocline (previously described by Rudels et al., 1996 and Steele & Boyd, 1998) that is distinct from a permanent halocline. This seasonal halocline temporarily separates the surface mixed layer from the thermocline but is eroded during winter mixing until additional stratification (supplied by relatively fresh Siberian shelf waters) restricts this mixing to shallower layers. The seasonal halocline transitions to a more permanent halocline across the front observed north of Severnaya Zemlya.**

**Sections 4.1 and 4.2 have been expanded in the revised manuscript and further clarify this distinction (e.g., Page 8, Lines 11-13).**

Page 4, line 9: In the supplementary material the depth of the winter mixed layer with salinity >34 ranges between 30m to 94m (L1) and between 42m and 58m (L2). Most of the depths were larger than 50m. (There were also two stations with depths close to 200m on L2 but the temperatures were close to 0C and the salinities >34.8. Should these observations be correct we would have winter mixing into the Atlantic water. This shows that the temperature minimum as a limit for winter convection should be used with care. However, in general it underestimates the winter mixed layer depth.)

**The stations exhibiting deeper winter mixed layers were located on the shelf (water depths ≤ 500 m) and were associated with cold and relatively homogeneous bottom layers as well as maximum Atlantic layer temperatures of < 0.5°C. We have added text to the table caption in the Supplementary Materials discussing potential errors in the assignment of winter mixed layer depths based upon this method. We have also visually inspected all of the potential temperature profiles and have removed WML depths from the table that either appear to be in error or seem ambiguous. We also note that these removals do not largely alter the mean WML depth or**

**salinity and therefore do not greatly impact our subsequent calculations or associated interpretations.**

Page 5, lines 19-21: Freezing shifts the mixing line between Atlantic water and meteoric water to the right, making it steeper. The Laptev Sea shelf input would then contain more brine than the East Siberian Sea input.

**That is correct. The overall dominance of brine over sea ice melt (i.e., net ice formation) in the Laptev Sea and the higher contribution of sea-ice meltwater in the East Siberian Sea versus the Laptev Sea have been documented in previous studies (e.g., Bauch et al., 2011; 2013; Anderson et al., 2013).**

**Text explaining the impact of ice formation on salinity and d18O has been added to the revised manuscript on Page 6, Lines 25-32.**

**We have also included citations to Bauch et al. (2011; 2013) and Anderson et al. (2013) on Page 7, Lines 9-15.**

Page 5, last paragraph: Here I am not sure that I follow the authors thinking. As the water from the Nansen Basin (the sea ice melt water branch) becomes covered by les saline shelf water with larger content of meteoric water, vertical mixing with the overlying water will lead to a mixing line with the observed slope. However, the bulk of the halocline water mass is derived from the winter mixed layer advected towards the Laptev Sea. The mixing changes slightly the properties of the halocline, but it does not provide any significant volume. Is this what the authors mean, or do they claim that here exits a major different source of the halocline water containing initially more meteoric water?

**We are not claiming that there exists a different source of halocline water. The reviewer's first interpretation is correct. Mixing with overlying, less saline waters results in small changes to salinity and d18O; however, these small changes initiate a movement from the SIM mixing relationship (prevalent on the western side of the defined front) to the MW mixing relationship (prevalent on the eastern side of the defined front) in salinity-d18O space that also corresponds with the migration of the θ-S "kink" (or "bend") that has typically been used to identify lower halocline water. This is the first step in the process that transforms the seasonal halocline into the permanent halocline and begins to form the cold halocline layer.**

**We have significantly expanded the text of sections 4.1 and 4.2 in the revised manuscript to clarify these points (e.g., Page 8, Lines 14-31).**

Page 6, lines 10-14: Once the low salinity polar mixed layer is formed, any shelf contribution having higher salinity than the polar mixed layer will contribute to the halocline. The question is, how large are these contributions compared with the initial Fram Strait branch and Barents Sea branch contributions?

**The reviewer makes an excellent point and we certainly agree that any shelf contribution with a salinity exceeding that of the freshened polar mixed layer will contribute to the halocline. Our observations suggest that the majority of these shelf contributions will occur eastward of the observed front north of Severnaya Zemlya and that initial contributions serve to cap LHW (establishing a permanent halocline) and further contributions will build the cold halocline layer. Thus, we argue LHW (34.2 ≤ S ≤ 34.5) is primarily "*basin-derived*" and the capping of this water mass by Siberian shelf waters both isolates LHW (completing the formation mechanism) and in so doing, forms the cold halocline layer. The majority of shelf waters contributing to the halocline will have a salinity < 34.2 and therefore contribute to the "lower MW mixing branch" defined in this study. While there are certainly exceptions, such as more saline waters formed in polynyas, we suggest that the mechanisms we describe are responsible for the bulk of halocline layer formation. We have noted in this paragraph that this hypothesis does not agree with the circulation scheme recently proposed by Bauch et al. (2016).**

**We have added text to the revised manuscript (Page 9, Lines 18-23) to clarify these points.**

Page 6, lines 26-28: This description is essentially correct, but why not state explicitly that to create an halocline water mass from the winter mixed layer, this layer has to be capped by a water mass with lower salinity advected from the Laptev Sea shelf?

**We have made this statement in the revised manuscript as suggested by the reviewer on Page 8, Lines 10-11.**

Page 6, lines 30-32, Page 7, lines 1-5: These mixing examples are interesting, but I think that it would be simpler to think of the winter mixed layer being capped by low salinity water. The initial thermostad and halostad would then be freshened by mixing with the low salinity layer above and heated and become more saline by mixing with the Atlantic water below. This would create the vertical gradients in temperature and salinity that characterize the halocline. Furthermore, I thought that freezing does not fractionate the oxygen isotopes. Brine rejection would then not change the oxygen-18 value. What is the reference for the adopted slope? On page 8, line 13 brine rejection is characterized as "negative sea ice melt". Would that not give a slope that increases the delta-oxygen-18 value as well as the salinity?

**We included this simple mixing scenario to further test the possibility of our proposed mechanism to explain both the d18O and potential temperature observations in the LHW. While we do not claim that this simple mixing is necessarily responsible for the observed halocline water properties, we note that such mixing can explain our observations.**

**We make this statement in the revised manuscript on Page 9, Lines 12-14.**

Ice formation (freezing) results in the rejection of salts from the sea ice matrix as well as a preferential rejection of 16O (the lighter isotope). As a result brine is characterized by higher salinities and a more negative d18O value whereas sea ice (and therefore sea ice meltwater) is characterized by a somewhat more positive d18O value. This fractionation is not large; fractionation factors range between about 1.6 and 2.8 ‰ depending upon the age of the ice and the rate of freezing (Eicken et al., 1998; Macdonald et al., 1995; Melling and Moore, 1995), but it does result in a steeper salinity-d18O slope (as illustrated in Fig. 5e of the revised manuscript).

Text has been added on Page 6, Lines 25-32 clarifying this point.

Page 7, lines 6-7: To me the lower halocline water and the cold halocline layer are the same, at least in this location. Once we enter the Canada Basin with Pacific inflow through Bering Strait the situation is different.

We somewhat disagree. We have argued in the introduction that the lower halocline water is a separate water mass that essentially marks the base of the cold halocline layer (formed from mixing and additional inputs from the shelves) and represents a transition between the halocline and reverse thermocline.

Also, see our response to the comment regarding Page 6, lines 10-14.

Page 7, lines 23-24: I agree with this statement.

Excellent.

Page 8, lines 10-11: Why does the delta-oxygen-18 decreases? A reference is needed.

The d18O decreases (along with salinity) because these waters mix with overlying waters characterized by lower salinities and lower d18O values, due to the influence from both river runoff (characterized by d18O values < -18 ‰) and brine (highly negative d18O values). We have referred to Fig. 5e at the end of this sentence in the revised manuscript (Page 11, Line 15) to illustrate these changes.

We also note that a full description of the changes in d18O due to mixing with MW and ice formation, complete with references, has been added in response to a previous comment (see Page 6, Lines 25-32 of the revised manuscript).

Page 8, lines 17-21: Does this mean that there are two separate branches of halocline water, or only that when the Fram Strait and the Barents branches cross the front, their upper parts are changed from being homogenized by brine rejection and convection to evolve by mixing with overlying low salinity and underlying Atlantic water?

We argue that the majority of our observations describe the formation and modification of a single source of lower halocline water (basin-derived or Fram

**Strait branch LHW) but note that we only glimpsed influences from Barents Sea Branch halocline water.**

**We have added text throughout the manuscript, but particularly on Page 11, Lines 9-10, pointing out this distinction.**

**We suggest, like Rudels et al. (1996), that the Fram Strait branch halocline water is changed from a seasonal halocline to a permanent halocline after crossing the Severnaya Zemlya front. The salinity-d18O characteristics are altered (transition from SIM to MW mixing branch) due to homogenization and ice formation and then the LHW is capped by Siberian shelf water east of the front, making the halocline permanent and building up the cold halocline layer.**

**We have also noted that this scheme contrasts with that recently published by Bauch et al. (2016).**

Figure 1: I would have appreciated to see not only TS curves but also profiles from the different sections.

**Figure 1 is rather large and we do not discuss features of the individual salinity and temperature profiles in the text. Instead, we have relied primarily on descriptions in θ-S space. While we do not agree that these profiles are needed in the main text, we have included two additional figures in the Supplementary Materials that show potential temperature and salinity profiles from selected stations along each transect so other interested readers may view them as desired.**

Summary: My concern about this manuscript should be clear after the comments above. If the authors mean that there are two distinct branches, one sea ice melt branch and one meteoric water branch, they have to argue their case better. If it is only a transition between the two branches across the front, the manuscript does not bring much new information. However, the use of the oxygen-18 data is interesting, but some of the adopted slopes have to be explained better.

**As stated in our responses above, we do not argue for two distinct branches but instead offer geochemical evidence from salinity and stable oxygen isotope mixing relationships to support a single mechanism that describes the transition from a seasonal to permanent halocline layer, in general agreement with the mechanism proposed by Rudels et al. (1996). While this evidence does not necessarily offer up a new mechanism by which to classify halocline water formation it does offer geochemical data that agrees with hypotheses based primarily on interpretations of θ-S data. We note that our conclusions also disagree with recent work published by Bauch et al. (2016) supporting distinct sources of lower halocline water from the Kara Sea shelf. Finally, the salinity-d18O mixing regimes presented in this study may be used in future studies to evaluate subsequent mixing and modification to the halocline layer. The apparent robust nature of these mixing regimes suggests that the "convective mechanism" of LHW formation has been more-or-less consistent**

**despite variations in the distribution and strength of the halocline layer. We argue that the stability of these relationships may make them suitable for the evaluation of modifications to the halocline due to mixing and/or interactions with shelf sediments that increase nutrient concentrations and decrease dissolved oxygen concentrations.**

The manuscript might be published after major revisions, some based on my comments given above.

**Based on the comments provided by the two reviewers, we have made major revisions that have improved the quality and clarity of the manuscript.**

[revised manuscript text omitted]

While mixing down to the previous year's WML (or shallower) might be expected given the increase in freshwater inventories (and stratification) moving from west to east along the slope, deeper mixing was observed in the study region between 2013 and 2015 (Polyakov et al., 2017). The depth of the 34.4 isohaline ranged between 60 and 100 m at the MW branch stations. If we consider mixing down to 100 m and 1 m of ice formation, the resulting salinity (34.50) and $\delta^{18}$O (0.07 ‰) resemble the upper MW branch at the break point. Thus, both shallower (~60 m) and deeper (~100 m) mixing result in a transition from the SIM branch to the MW branch. Although mixing and brine release can account for salinity and $\delta^{18}$O changes, additional mixing (either lateral or vertical) with warm AW is needed to produce the $\theta \approx$ -1 °C that is associated with the LHW of the MW branch. A mixture comprising ~79 % of newly formed MW branch water (34.38, -0.08 ‰, and -1.89 °C) and ~21 % AW (34.9, 0.3 ‰, and 2 °C) would produce the salinity (34.4), $\theta$ (-1.07 °C), and $\delta^{18}$O (0 ‰) observed. We have included this simple mixing scenario to further test the possibility of our proposed mechanism to explain both the $\delta^{18}$O and potential temperature observations in the LHW. While we do not claim that this simple mixing is necessarily responsible for the observed halocline water properties, we note that such mixing can explain our observations.

It is also important to note that MW must have been supplied to the region north of SZ to define the front separating SIM and MW branches. We adopt the suggestion made by Bauch et al. (2016) that waters moving off the shelf in the northeastern Laptev Sea (i.e., along the Lomonosov Ridge) are recirculated westward, except we suggest this recirculation does not necessarily provide four distinct sources of halocline water. Any shelf contribution with a salinity exceeding that of the relatively fresh polar mixed layer will contribute to the halocline. Our observations suggest that the majority of these shelf contributions will occur eastward of the SZ front. We argue that LHW (34.2 < S < 34.5) is primarily *basin-derived* and initial shelf water contributions serve to cap LHW (and begin to establish the permanent halocline) whereas further contributions to the halocline will have a salinity < 34.2 and therefore contribute to the "lower MW mixing branch" and build the CHL. In support of this hypothesis, we note that the salinity and $\delta^{18}$O values characterizing the four LHW types defined by Bauch et al. (2016) form a salinity-$\delta^{18}$O mixing line ($\delta^{18}$O = 0.9828S – 33.901) similar to the lower MW branch identified in this study (Supplementary Figure S2). This could indicate that the four LHW types described by Bauch et al. (2016) are actually mixtures of *basin-derived* LHW and increasing contributions of MW progressing eastward from SZ.

**4.3 Stability of $\delta^{18}$O-S mixing regimes**

Using salinity and $\delta^{18}$O observations, we have outlined a hypothesis to describe the transition from a seasonal halocline, formed due to mixing between AW and SIM west of SZ, to a permanent halocline involving winter mixing, ice formation, and the introduction of Siberian shelf waters characterized by high influences of MW and brine east of SZ that largely follows the hypothesis previously described by Rudels et al. (1996). However, we have thus far relied upon data collected

Matthew Alkire 9/18/17 11:50 AM

Matthew Alkire 9/5/17 9:09 PM

Matthew Alkire 9/15/17 1:41 PM
Formatted [... [37]]

Matthew Alkire 9/15/17 1:41 PM

Matthew Alkire 9/18/17 12:01 PM
Formatted [... [38]]

Matthew Alkire 9/18/17 11:56 AM

Matthew Alkire 9/15/17 1:05 PM

Matthew Alkire 9/15/17 1:06 PM

Matthew Alkire 9/14/17 12:00 PM
Formatted [... [39]]

Matthew Alkire 9/18/17 12:12 PM

[revised manuscript text omitted]
. We note that the identification of the WML depth by this method is associated with some uncertainty and may be particularly ambiguous at stations with a mixed layer close to the freezing point. The WML depths estimated using this method were visually checked against vertical profiles of potential temperature and θ-S diagrams. Stations that appeared to have no clearly identifiable θ$_{min}$ or multiple minima are marked with "*CND*" (could not determine). "Salt mixed" and "θ mixed" refer to the mean salinities and potential temperatures estimated from individual profiles assuming the water column will be homogenized down to the previous year's WML.

| Station | Transect | WML Depth (m) | WML Salinity | WML θ | Salt mixed | θ mixed |
|---|---|---|---|---|---|---|
| 1 | - | 52 | 34.556 | -0.699 | 34.112 | 1.0 |
| 2 | - | 56 | 34.368 | -1.674 | 33.867 | -1.3 |
| 3 | - | 45 | 34.365 | -1.384 | 33.876 | -1.2 |
| 4 | - | 52 | 34.414 | -1.533 | 33.849 | -1.5 |
| 5 | - | 49 | 34.443 | -1.485 | 33.563 | -1.2 |
| 6 | - | 48 | 34.236 | -1.663 | 32.940 | -0.4 |
| 7 | L5 | *CND* | - | - | - | - |
| 8 | L5 | *CND* | - | - | - | - |
| 9 | L5 | 85 | 34.282 | -1.659 | 33.084 | -0.4 |
| 10 | L5 | 87 | 34.317 | -1.611 | 33.350 | -0.6 |
| 11 | L5 | 58 | 34.024 | -1.750 | 32.457 | -0.5 |
| 12 | L5 | 50 | 33.893 | -1.792 | 32.896 | -1.0 |
| 13 | L5 | 64 | 33.896 | -1.799 | 32.744 | -1.2 |
| 14 | L5 | 56 | 33.918 | -1.797 | 32.947 | -1.4 |
| 15 | L5 | 52 | 34.023 | -1.775 | 33.315 | -1.6 |
| 16 | L5 | 67 | 34.210 | -1.727 | 33.573 | -1.6 |
| 17 | L5 | 53 | 33.968 | -1.762 | 33.141 | -1.6 |
| 18 | L5 | 52 | 34.149 | -1.721 | 33.285 | -1.583 |
| 19 | L5 | 40 | 33.995 | -1.733 | 33.098 | -1.622 |
| 20 | L5 | 42 | 34.011 | -1.747 | 32.939 | -1.611 |
| 21 | L5 | 59 | 33.994 | -1.745 | 33.076 | -1.622 |
| 22 | L5 | 45 | 33.720 | -1.730 | 32.331 | -1.577 |
| 23 | L5 | 55 | 33.934 | -1.754 | 32.526 | -1.627 |
| 24 | L5 | 39 | 33.383 | -1.770 | 32.022 | -1.610 |
| 25 | L5 | 68 | 33.838 | -1.819 | 32.630 | -1.671 |
| 26 | L5 | 47 | 33.464 | -1.759 | 31.888 | -1.611 |
| 27 | - | *CND* | - | - | - | - |
| 28 | - | *CND* | - | - | - | - |
| 29 | L6 | 70 | 34.022 | -1.675 | 31.551 | -1.127 |
| 30 | L6 | 70 | 34.056 | -1.633 | 31.863 | -1.348 |
| 31 | L6 | 61 | 33.801 | -1.655 | 31.125 | -1.320 |
| 32 | L6 | 73 | 34.006 | -1.664 | 31.556 | -1.482 |

Matthew Alkire 9/15/17 11:12 AM
Deleted: **Table S1.** Linear regression analyses (restricted to salinities ≥ 34.5) of salinity-δ$^{18}$O measurements collected along transects occupied during the 2013. Slopes, intercepts, correlation coefficients (r) and associated standard errors (se) are reported for each transect as well as the collection of transects comprising the sea-ice melt (SIM) and meteoric (MW) water branches. **Transect** ... [1]

Matthew Alkire 9/15/17 11:11 AM

Matthew Alkire 9/5/17 8:15 PM

Matthew Alkire 9/5/17 8:16 PM

Matthew Alkire 9/5/17 8:16 PM

Matthew Alkire 9/5/17 8:17 PM

Matthew Alkire 9/5/17 8:44 PM

Matthew Alkire 9/5/17 2:59 PM

Matthew Alkire 9/5/17 2:59 PM

Matthew Alkire 9/5/17 3:01 PM

Matthew Alkire 9/5/17 8:46 PM
Formatted Table

Matthew Alkire 9/5/17 8:46 PM

Matthew Alkire 9/5/17 8:46 PM
Formatted Table

| | | | | | | |
|---|---|---|---|---|---|---|
| 33 | L6 | 51 | 33.513 | -1.717 | 31.162 | -1.434 |
| 34 | L6 | 46 | 32.992 | -1.675 | 30.677 | -1.489 |
| 35 | L6 | 40 | 32.340 | -1.620 | 30.412 | -1.468 |
| 36 | L6 | 44 | 33.242 | -1.739 | 30.896 | -1.532 |
| 37 | L6 | 46 | 32.973 | -1.731 | 31.124 | -1.603 |
| 38 | L6 | 75 | 34.020 | -1.706 | 32.116 | -1.597 |
| 39 | - | 59 | 33.150 | -1.753 | 32.020 | -1.659 |
| 40 | - | 44 | 32.557 | -1.673 | 30.467 | -1.254 |
| 41 | - | 46 | 32.484 | -1.742 | 31.099 | -1.620 |
| 42 | - | 39 | 32.494 | -1.712 | 31.246 | -1.651 |
| 43 | - | 37 | 32.202 | -1.679 | 31.044 | -1.652 |
| 44 | - | 52 | 32.384 | -1.693 | 31.357 | -1.668 |
| 45 | L5.5 | 43 | 33.180 | -1.600 | 30.802 | -0.745 |
| 46 | L5.5 | 51 | 33.897 | -1.659 | 31.681 | -0.435 |
| 47 | L5.5 | CND | - | - | - | - |
| 48 | L5.5 | 48 | 33.980 | -1.683 | 32.122 | -0.542 |
| 49 | L5.5 | 63 | 34.234 | -1.571 | 32.615 | -0.603 |
| 50 | L5.5 | 58 | 34.204 | -1.591 | 32.590 | -0.622 |
| 51 | L5.5 | CND | - | - | - | - |
| 52 | L5.5 | CND | - | - | - | - |
| 53 | L5.5 | CND | - | - | - | - |
| 54 | L5.5 | CND | - | - | - | - |
| 55 | L5.5 | CND | - | - | - | - |
| 56 | L5.5 | 50 | 33.867 | -1.776 | 32.323 | -1.124 |
| 57 | L5.5 | 59 | 33.921 | -1.774 | 32.931 | -1.100 |
| 58 | L5.5 | 63 | 33.961 | -1.766 | 32.750 | -0.674 |
| 59 | - | 65 | 34.051 | -1.728 | 33.112 | -0.711 |
| 60 | L5 | 62 | 33.986 | -1.762 | 32.831 | -0.674 |
| 61 | L5 | 44 | 33.991 | -1.728 | 33.177 | -1.271 |
| 62 | L5 | 44 | 34.000 | -1.737 | 33.045 | -1.000 |
| 63 | L4 | CND | - | - | - | - |
| 64 | L4 | CND | - | - | - | - |
| 65 | L4 | 26 | 33.959 | -1.670 | 32.814 | -1.067 |
| 66 | L4 | CND | - | - | - | - |
| 67 | L4 | 42 | 34.297 | -1.688 | 33.673 | -1.421 |
| 68 | L4 | 51 | 34.334 | -1.689 | 33.783 | -1.381 |
| 69 | L4 | 46 | 34.353 | -1.716 | 33.739 | -1.404 |
| 70 | L3 | 20 | 32.257 | -1.718 | 32.246 | -1.714 |
| 71 | L3 | 15 | 31.864 | -1.718 | 31.864 | -1.719 |
| 72 | L3 | CND | - | - | - | - |
| 73 | L3 | CND | - | - | - | - |
| 74 | L3 | 31 | 33.955 | -1.686 | 33.097 | -1.588 |
| 75 | L3 | 22 | 34.040 | -1.683 | 33.327 | -1.517 |

| | | | | | | |
|---|---|---|---|---|---|---|
| 76 | L3 | 33 | 34.195 | -1.663 | 33.208 | -1.602 |
| 77 | L3 | 48 | 34.322 | -1.681 | 33.715 | -1.560 |
| 78 | L3 | 52 | 34.354 | -1.610 | 33.692 | -1.441 |
| 79 | L3 | CND | - | - | - | - |
| 80 | L3 | CND | - | - | - | - |
| 81 | L3 | CND | - | - | - | - |
| 82 | L2 | CND | - | - | - | - |
| 83 | L2 | CND | - | - | - | - |
| 84 | L2 | CND | - | - | - | - |
| 85 | L2 | 54 | 34.309 | -1.385 | 33.709 | -0.538 |
| 86 | L2 | 53 | 34.336 | -1.566 | 33.798 | -0.783 |
| 87 | L2 | 42 | 34.370 | -1.545 | 33.711 | -0.567 |
| 88 | L2 | 55 | 34.360 | -1.584 | 33.799 | -0.664 |
| 89 | L2 | 51 | 34.392 | -1.623 | 33.825 | -0.995 |
| 90 | L2 | 52 | 34.390 | -1.694 | 33.893 | -1.255 |
| 91 | L2 | 58 | 34.405 | -1.723 | 33.800 | -1.410 |
| 92 | L2 | CND | - | - | - | - |
| 93 | L2 | 51 | 34.393 | -1.648 | 33.823 | -1.166 |
| 94 | - | 87 | 34.420 | -1.770 | 34.014 | -1.624 |
| 95 | - | 44 | 33.993 | -1.727 | 32.937 | -1.689 |
| 96 | - | CND | - | - | - | - |
| 97 | L1 | CND | - | - | - | - |
| 98 | L1 | CND | - | - | - | - |
| 99 | L1 | 85 | 34.366 | -1.795 | 33.925 | -1.647 |
| 100 | L1 | CND | - | - | - | - |
| 101 | L1 | 71 | 34.370 | -1.764 | 34.050 | -1.601 |
| 102 | L1 | 58 | 34.367 | -1.719 | 34.004 | -1.460 |
| 103 | L1 | 58 | 34.372 | -1.710 | 33.861 | -1.332 |
| 104 | L1 | 66 | 34.390 | -1.705 | 33.995 | -1.409 |
| 105 | L1 | 59 | 34.379 | -1.728 | 33.942 | -1.337 |
| 106 | L1 | 30 | 34.363 | -1.547 | 33.654 | -0.659 |
| 107 | L1 | 45 | 34.289 | -1.610 | 33.781 | -0.443 |
| 108 | L1 | 45 | 34.292 | -1.589 | 33.737 | -0.198 |
| 109 | SAT | CND | - | - | - | - |
| 110 | SAT | CND | - | - | - | - |
| 111 | SAT | 48 | 34.626 | -0.168 | 34.309 | 1.795 |
| 112 | SAT | 43 | 34.583 | -0.213 | 34.245 | 1.788 |
| 113 | SAT | 42 | 34.526 | -0.535 | 34.010 | 1.085 |
| 114 | SAT | 38 | 34.337 | -1.411 | 33.769 | -0.176 |
| 115 | SAT | 35 | 34.274 | -1.355 | 33.708 | 0.080 |
| 116 | SAT | 39 | 34.322 | -1.321 | 33.626 | 0.141 |

-
**Averages**

[revised manuscript text omitted]

**Figure S3**. Vertical profiles of potential temperature (θ) and salinity plotted as blue and red lines, respectively, for selected stations on the SAT, L1, L2, and L3 transects. Stations were selected that generally represented the hydrographic conditions observed nearest the continental shelves ("onshore"), on the slope ("transitional"), and in the deep basins (offshore) along each transect. Note that, while the temperature and salinity axes are identical among panels, the range of the y-axes (depth) varies with each panel.

Matthew Alkire 9/18/17 1:46 PM

[Figure]

**Figure S4**. Vertical profiles of potential temperature (θ) and salinity plotted as blue and red lines, respectively, for selected stations on the L4, L5, L5.5, and L6 transects. Stations were selected that generally represented the hydrographic conditions observed nearest the continental shelves ("onshore"), on the slope ("transitional"), and in the deep basins (offshore) along each transect. Note that, while the temperature and salinity axes are identical among panels, the range of the y-axes (depth) varies with each panel.

---

## Author Response (AR2)

**Combining physical and geochemical methods to investigate lower halocline water formation and modification along the Siberian continental slope**

Matthew B. Alkire[1], Igor Polyakov[2], Robert Rember[2], Andrey Pnyushkov[2], Vladimir Ivanov[3,2], Igor Ashik[3]

[1]Applied Physics Laboratory, University of Washington, Seattle, WA USA
[2]International Arctic Research Center, University of Alaska Fairbanks, Fairbanks, AK USA
[3]Arctic and Antarctic Research Institute, St. Petersburg, RUS

*Correspondence to*: Matthew B. Alkire (malkire@apl.washington.edu)

All technical corrections required by the editor and reviewer have been implemented in the corrected manuscript.